# KNOWLEDGE-INFUSED PROMPTING: ASSESSING AND ADVANCING CLINICAL TEXT DATA GENERATION WITH LARGE LANGUAGE MODELS

## ABSTRACT

Clinical natural language processing requires methods that can address domain-specific challenges, such as complex medical terminology and clinical contexts. Recently, large language models (LLMs) have shown promise in this domain. Yet, their direct deployment can lead to privacy issues and are constrained by resources. To address this challenge, we delve into synthetic clinical text generation using LLMs for clinical NLP tasks. We propose an innovative, resource-efficient approach, CLINGEN, which infuses knowledge into the process. Our model involves clinical knowledge extraction and context-informed LLM prompting. Both clinical topics and writing styles are drawn from external domain-specific knowledge graphs and LLMs to guide data generation. Our extensive empirical study across 7 clinical NLP tasks and 16 datasets reveals that CLINGEN consistently enhances performance across various tasks, effectively aligning the distribution of real datasets and significantly enriching the diversity of generated training instances. We will publish our code and all the generated data upon acceptance.

## 1 INTRODUCTION

Clinical Natural Language Processing (NLP) emerges as a distinct subfield including the extraction, analysis, and interpretation of medical data from unstructured clinical text (Wornow et al., 2023). Despite its significance, unique challenges evolve for methodology development in clinical NLP. For example, clinical texts are often dense with abbreviations and specialized medical terminologies that can be perplexing to standard NLP models (Lee et al., 2023). Fortunately, recent advances in Large Language Models (LLMs) (Brown et al., 2020; Chung et al., 2022; Ouyang et al., 2022; OpenAI, 2023a;b) provide a promising way to resolve these issues, as they contain billions of parameters and have been pretrained on massive corpora, thus inherently capture a significant amount of clinical knowledge (Agrawal et al., 2022; Nori et al., 2023; Eric et al., 2023; Wong et al., 2023; Singhal et al., 2023a;b; Luo et al., 2022; Liu et al., 2023b). These progresses inspire the need for designing specialized approaches for adapting LLMs to clinical settings, which both address the terminology complexities and improve models through clinical data finetuning (Tu et al., 2023; Liu et al., 2023a).

Despite the strong capacity of general LLMs, directly applying them to infer over clinical text data is often undesired in practice. Firstly, these LLMs often have billions of parameters that translate to significant computational resources even for inference, leading to *increased infrastructure costs* and *long inference time*. Furthermore, the sensitive patient information contained in the clinical text naturally raises *privacy and regulatory compliance concerns* (Meskó & Topol, 2023; Keeling, 2023). To effectively combat these challenges, generating synthetic training data using LLMs stands out as a promising solution: It leverages the capability of LLMs in a way that is both resource-efficient and privacy-centric. When trained with these synthetic datasets mimicking real-world clinical data, models can achieve high performance while obeying data protection regulations.

Synthetic data generation with foundation models is a popular research domain in general machine learning (Azizi et al., 2023; Borisov et al., 2023; Meng et al., 2022; Ye et al., 2022a;b). However, when considering producing high-quality data that conforms to the distribution of the original dataset, simply adapting LLMs trained on general texts to generate clinical data presents unique challenges. To assess the quality of data generated by current methods, we carry out an exhaustive evaluation centered on distribution and diversity, detailed in Section 3. Insights from the t-SNE

embeddings visualization and the Central Moment Discrepancy (CMD) score indicate a noteworthy data distribution shift. We further examine the clinically-related entity quantities and frequencies in synthetic data, where a notable decline is observed when contrasting synthetic data with ground truth data. While some research has delved into clinical data generation with language models, many of these efforts are tailored to specific tasks. Examples include medical dialogues (Chintagunta et al., 2021), clinical notes (Giorgi et al., 2023), medical text mining (Tang et al., 2023), and electronic health records (Ive et al., 2020; Wang & Sun, 2022; Theodorou et al., 2023; Qian et al., 2023). These studies often directly adopt language models for text generation, and sometimes on excessive training data. Till now, a unified principle to better adapt LLMs for generating synthetic text for facilitating clinical downstream applications is still missing.

Motivated by the above analysis, we propose CLINGEN, a *clinical knowledge-infused* generic framework for high-quality clinical text generation in few-shot scenarios. Our ultimate goal is to narrow the gap between synthetic and ground-truth data and encourage the topic diversity of the generated text. Towards this end, we propose a strategy to utilize clinical knowledge extraction to contextualize the prompts. This includes generating clinical topics on entity and relation information from both KGs and LLMs and deriving writing style suggestions from LLMs. By doing this, CLINGEN integrates both non-parametric insights from external clinical knowledge graphs with the intrinsic parametric knowledge encoded in LLMs. It is worth noting that, CLINGEN only rely on minimal additional human efforts, and can be readily applied to a wide array of core tasks in clinical NLP.

Our contributions can be summarized as follows:

- We propose CLINGEN, a generic clinical knowledge-infused framework for clinical text data generation in few-shot settings. It can be readily applied to a wide range of tasks in clinical NLP.
- We present a simple yet effective strategy to utilize clinical knowledge extraction to customize the prompts toward target clinical NLP tasks. This includes generating clinical topics from both KGs and LLMs and deriving writing style suggestions from LLMs.
- We conduct an exhaustive evaluation of synthetic clinical data generation **across 7 clinical NLP tasks and 16 datasets**. Empirical findings demonstrate that CLINGEN not only aligns more closely with the distribution of the original data but also amplifies the diversity of the generated training samples. The empirical performance gains are consistent across various tasks with different LLMs and classifiers (8.98% for PubMedBERT$_{\text{Base}}$ and 7.27% for PubMedBERT$_{\text{Large}}$).

## 2 RELATED WORK

Generating additional training data enables a more precise analysis of medical text, and has gained more attention in the past years. Earlier research has employed data augmentation techniques to generate similar samples to existing instances with word substitution (Kang et al., 2021; Ribeiro et al., 2020), back translation (Xie et al., 2020), pretrained transformers (Kumar et al., 2020; Zhou et al., 2021; 2022) for enhancing model generalization. But they often yield rigid transformations and the quality of the augmented text cannot be always guaranteed. Another line of work focuses on leveraging external knowledge to create weak labels (Ratner et al., 2017; Fries et al., 2017; Wang et al., 2019; Dunnmon et al., 2020). These methods typically require domain expertise and additional task-specific corpora, which can be resource-intensive to obtain for low-resource clinical tasks.

The emergence of LLMs has presented new possibilities, and some studies attempt to use LLM to generate training data (Meng et al., 2022; 2023; Ye et al., 2022a; Yu et al., 2023; Chung et al., 2023), often with few demonstrations (Yoo et al., 2021). However, these methods often use generic and simple prompts that may not fully capture domain-specific knowledge, thus potentially limiting the quality of the generated data. Liu et al. (2022a); Chung et al. (2023) employ interactive learning to generate additional instances to refine the existing dataset, at the cost of additional human efforts. One recent study (Tang et al., 2023) explores synthetic data generation for clinical NLP. Nevertheless, their proposed approach relies on a *much larger training set* to generate candidate entities, which disregards the practical low-resource setting (Perez et al., 2021). Furthermore, their study is limited to a narrow range of tasks (2 tasks and 4 datasets only), lacking breadth in terms of exploring a diverse set of clinical NLP applications.

On the other hand, several works aimed at optimizing prompts using LLM itself (Mishra et al., 2022; Zhou et al., 2023b; Yang et al., 2023) or knowledge graphs (Chen et al., 2022b; Hu et al., 2022; Liu

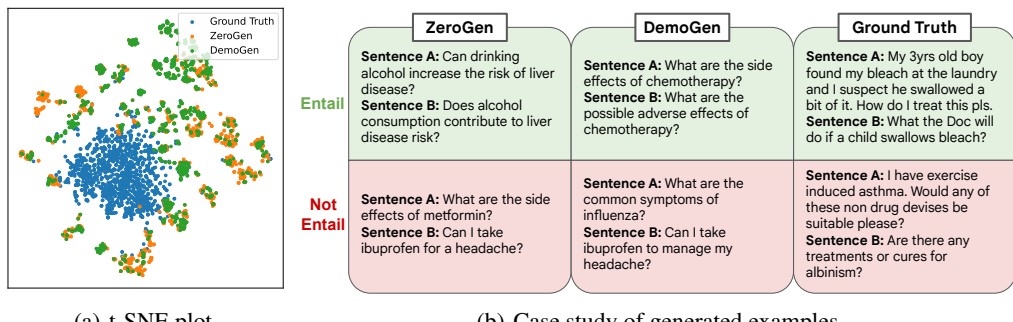

(a) t-SNE plot        (b) Case study of generated examples

Figure 1: Preliminary Studies. (a) is from BC5CDR-Disease and (b) is from MEDIQA-RQE.

et al., 2022b), yet they mainly focus on refining prompts to obtain the answer for the given input, and the prompt template often remains unchanged. Instead, we focus on the different task of generating training instances. By composing different topics and styles together, we are able to generate diverse templates for prompting LLMs to improve the quality of the synthetic data.

## 3 PRELIMINARY STUDY

This section first presents the foundational setup of synthetic data generation. Then, we provide an in-depth investigation into the pitfalls of existing synthetic data generation methods.

### 3.1 PROBLEM SETUP

In this paper, we study the synthetic data generation problem in the few-shot setting. The input consists of a training set $\mathcal{D}_{train} = \{(x_i, y_i)\}_{i=1}^{K}$, where each $(x_i, y_i)$ pair represents an input text and its corresponding label for the $i$-th example. $K$ denotes the total number of training samples, which is intentionally kept at a very small value (5-shot per label). The primary objective is to harness the capabilities of an LLM $\mathcal{M}$ to generate a synthetic dataset, denoted as $\mathcal{D}_{syn} = \{(\widetilde{x}_i, \widetilde{y}_i)\}_{i=1}^{N}$, where $N$ is the number of generated samples ($N \gg K$). For each downstream task, we fine-tune a classifier $\mathcal{C}_\theta$ (a moderate-size pre-trained language model) parameterized by $\theta$ on the synthetic dataset $\mathcal{D}_{syn}$ for evaluating on the target task[1].

### 3.2 LIMITATIONS OF EXISTING SYNTHETIC DATA GENERATION METHODS

Here, we take a closer look at the synthetic text data generated by two representative approaches: ZeroGen (Ye et al., 2022a), which directly instructs LLMs for data generation, and DemoGen (Yoo et al., 2021; Meng et al., 2023), which augments the prompt with few-shot demonstrations. We observe that these methods often introduce distribution shifts and exhibit limited diversity, which can be suboptimal for improving downstream performance. The illustration is as follows, and we include additional figures in Appendix B.

**Distribution Shift.** An inherent challenge when adapting LLMs to specific domains for text generation is the issue of *distribution shift*, given that LLMs are primarily trained on vast amounts of web text in general domains. In Figure 1(a), we visualize the embeddings[2] of both the ground truth training data and synthetic datasets generated via two representative methods. Overall, these methods use generic prompts (see Appendix F.3 for details) with minimal domain-specific constraints. This limitation remains evident even when incorporating few-shot demonstrations into the process, with a notable disparity between the embeddings of the ground truth data and synthetic data.

To quantify the data distribution shift, we employ Central Moment Discrepancy (CMD) (Zellinger et al., 2017) to measure the gap between synthetic and real data across six clinical NLP datasets. Particularly, a high CMD value indicates a large gap between the two given distributions. Figure 2(a) illustrates that both ZeroGen and DemoGen exhibit elevated CMD scores, indicating substantial dissimilarity between the synthetic data and those of the real dataset.

---

[1]While In-context Learning (Brown et al., 2020) can also be utilized, it is often hard to fit all generated instances into the context window, especially for datasets with high cardinality.

[2]We employ SentenceBERT (Reimers & Gurevych, 2019) as the text encoder.

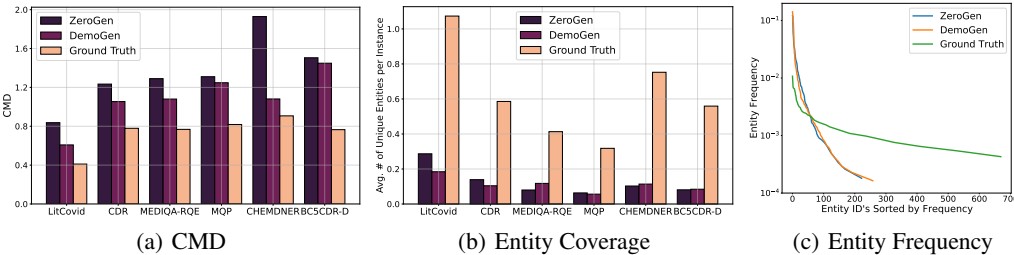

Figure 2: Preliminary Studies. (c) is from BC5CDR-Disease and is in log scale.

**Limited Diversity.** Clinical datasets in real-world scenarios harbor a wealth of valuable knowledge that can be challenging to replicate within synthetically generated data by AI models. We evaluate synthetic dataset diversity by using both entity quantity and their normalized frequencies. The results are illustrated in Figures 2(b) and 2(c). Our analysis reveals that datasets generated by ZeroGen and DemoGen exhibit a limited number of clinical entities, having a substantial discrepancy with the ground truth. Furthermore, it is highlighted that only a minority of potential entities and relations are frequently referenced across instances, while the majority are generated infrequently.

To explicitly illustrate the aforementioned limitations of synthetic datasets created using existing methods, we present a case study in Figure 1(b). In this case study, we randomly select one sample from each class within the training set generated by ZeroGen and DemoGen. These selected samples are compared with the ground truth data from the MEDIQA-RQE dataset, which aims to predict whether a consumer health query can entail an existing Frequently Asked Question (FAQ). The comparison reveals that the samples generated by ZeroGen and DemoGen tend to be more straightforward, lacking the *sufficient details* and *real-case nuances* present in the ground truth data. Furthermore, the generated samples adhere to a more uniform style and structure, while the ground truth encompasses various situations and writing styles, including urgent and informal inquiries.

## 4 CLINICAL KNOWLEDGE INFUSED DATA GENERATION

The revealed insights from the preliminary studies assert the necessity of domain-tailored knowledge for clinical synthetic data generation. In pursuit of efficient, effective, and scalable data generation for clinical domains, we introduce our novel framework, CLINGEN, a prior knowledge-informed clinical data generation. The overview of CLINGEN is shown in Figure 3. This innovative two-step methodology harnesses the emergent capabilities of LLMs and external knowledge from KGs to facilitate the synthesis of clinical data, even when only presented with few-shot examples.

### 4.1 CLINICAL KNOWLEDGE EXTRACTION

Contrary to previous studies (Ye et al., 2022a;b; Meng et al., 2022; 2023) which employ generic queries to prompt LLMs for text generation, CLINGEN emphasizes refining clinically informed prompts. This approach aims to extract rich clinically relevant knowledge from parametric (*e.g.* LLMs) or nonparametric sources (*e.g.* knowledge graphs) and tailor it to clinical NLP tasks. To realize this objective, our modeling contains two dimensions including *clinical topics* and *writing styles*, which are integrated into the original prompts to infuse domain-specific knowledge. By dynamically composing different topics and writing styles together, CLINGEN can provide a diverse suite of prompts, resulting in a wider spectrum of text produced from LLM.

#### 4.1.1 CLINICAL TOPICS GENERATION

We provide two choices to generate clinical topics – one is to sample related entities or relations from external KG, and the other is to query relevant knowledge from LLM.

**Topics sampled from Non-Parametric KGs.** Healthcare KGs offer a rich collection of medical concepts and their complex relationships, and have emerged as a promising tool for organizing medical knowledge in a structured way (Li et al., 2022). In our methodology, we employ the iBKH KG (Su et al., 2023) due to its broad coverage over clinical entities. To illustrate, for the Disease Recognition task (NCBI) (Dogan et al., 2014), we extract all medication nodes from the iBKH to

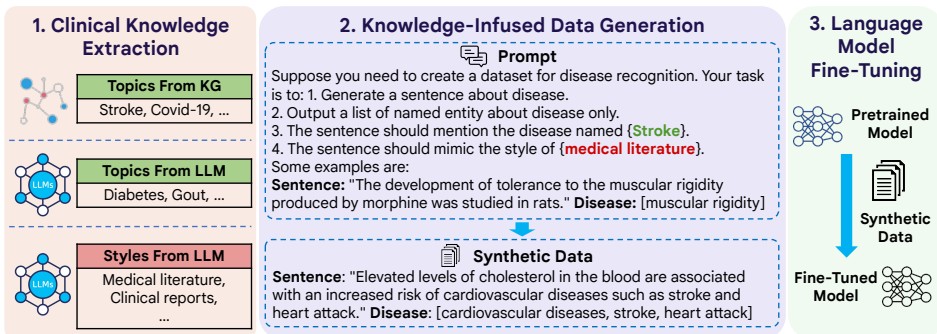

Figure 3: The overview of CLINGEN.

bolster the pharmaceutical information. As another example, we retrieve links between drug and disease nodes for the chemical and disease relation extraction (CDR) task (Wei et al., 2016). By integrating the information from the clinical KG into our data generation process, we guarantee that our generated samples exhibit a high degree of contextual accuracy, diversity, and semantic richness.

**Topics queried from Parametric Model (LLMs).** LLMs provide an alternative method for acquiring domain knowledge, as they are pre-trained on extensive text corpora, including medical literature. Specifically, we aim to harness the rich clinical domain knowledge encoded in ChatGPT (`gpt-3.5-turbo-0301`) to augment the prompt. The incorporated prior knowledge from LLMs is focused on entity categories that hold significant relevance within clinical text datasets, including diseases, drugs, symptoms, and side effects. For each of these pivotal entity types, we prompt the LLMs by formulating inquiries, *e.g.*, "`Suppose you are a clinician and want to collect a set of <Entity Type>. Could you list 100 entities about <Entity Type>?`". These crafted conversational cues serve as effective prompts, aiding in the retrieval of clinically significant entities from the extensive domain knowledge within LLMs. For each entity type, we generate 300 entities which will be used for synthetic data generation.

### 4.1.2 WRITING STYLES SUGGESTION

**Styles suggested by LLMs.** To address the limitations mentioned in Sec 3.2 and introduce a diverse range of writing styles into the generated samples, we leverage the powerful LLM again by suggesting candidate writing styles for each task. Specifically, we incorporate task names into our prompts (e.g., disease entity recognition, recognizing text entailment, etc.) and integrate few-shot demonstrations. We then engage ChatGPT in suggesting several potential sources, speakers, or authors of the sentences. See Appendix F.1 for detailed prompt. Responses such as "`medical literature`" or "`patient-doctor dialogues`" are augmented into the prompts to imitate the writing styles found in real datasets.

### 4.2 KNOWLEDGE-INFUSED SYNTHETIC DATA GENERATION

With the generated entities as well as styles, the key challenge becomes how to leverage them to extract rich clinical information from the LLM for improving synthetic data quality. Directly putting all the elements to enrich the prompt is often infeasible due to the massive size of entities. To balance informativeness as well as diversity, we propose a knowledge-infused strategy, where the collected clinical topics and writing styles serve as the base unit. In each step, we randomly sample a topic and a writing style from the candidate set to augment the prompts. For instance, for the Disease Recognition (NCBI) task, consider a clinical entity like "`stroke`". We enrich the prompt query for LLM by appending "`generate a sentence about stroke`" as a generation guidance. For a comprehensive view of the prompt formats across various tasks, please refer to Appendix F. Despite its simplicity, this knowledge-infused strategy ensures that the clinical context is incorporated into the prompts while encouraging prompt diversity (via composing different entities and writing styles), thereby enhancing the quality and clinical relevance of the generated synthetic data.

### 4.3 LANGUAGE MODEL FINE-TUNING

After generating synthetic data $\mathcal{D}_{\text{syn}}$ through LLMs, we fine-tune a pre-trained classifier $\mathcal{C}_\theta$ to each downstream task. Following (Meng et al., 2023), we first fine-tune $\mathcal{C}_\theta$ on $\mathcal{D}_{\text{train}}$ with standard super-

vised training objectives (denoted as $\ell(\cdot)$), then on the synthetic data $\mathcal{D}_{\text{syn}}$ as

$$\text{Stage I} : \theta^{(1)} = \min_{\theta} \ \mathbb{E}_{(x,y)\sim\mathcal{D}_{\text{train}}} \ell\left(f(x;\theta), y\right), \tag{1}$$

$$\text{Stage II} : \theta^{(2)} = \min_{\theta} \ \mathbb{E}_{(x,y)\sim\mathcal{D}_{\text{syn}}} \ell\left(f(x;\theta), y\right), \theta_{\text{init}} = \theta^{(1)}. \tag{2}$$

It's important to highlight that we strictly follow a standard fine-tuning process and avoid using any extra techniques: (1) for standard classification tasks, $\ell(\cdot)$ is the cross-entropy loss; (2) for multi-label classification tasks, $\ell(\cdot)$ is the binary cross-entropy loss; (3) for token-level classification tasks, we stack an additional linear layer as the classification head and $\ell(\cdot)$ is the token-level cross-entropy loss. The design of *advanced learning objectives* as well as *data mixing strategies*, while important, are orthogonal to the scope of this paper.

## 5 EMPIRICAL EVALUATION

Given our focus on synthetic text generation, our primary interest lies in faithfully evaluating different synthetic text generation approaches under few-shot scenarios, rather than engaging in a "state-of-the-art" race with general few-shot learning approaches (*i.e.* we never claim that we achieve *state-of-the-art* performance on these tasks). In this context, the following questions particularly intrigue us: **RQ1**: How does CLINGEN perform when compared with baselines on different downstream tasks? **RQ2**: How do different factors such as LLM generators and the size of synthetic data affect the performance of CLINGEN? **RQ3**: How is the quality of the synthetic datasets generated by CLINGEN and baselines? These questions are addressed in Sec 5.2, Sec 5.3 and Sec 6, respectively.

### 5.1 EXPERIMENT SETUP

We conduct experiments in the few-shot settings with 5 examples available for each class. We employ ChatGPT (OpenAI, 2023a) (`gpt-3.5-turbo-0301`) as the generator and maintain the same amount of synthetic training data for both CLINGEN and baselines for a fair comparison. The pre-trained PubMedBERT (Gu et al., 2021) is then applied to fine-tune on the generated synthetic data for both CLINGEN and baselines, where we consider both the `Base` and `Large` variants. See Appendix C for implementation details.

**Datasets and Tasks.** In our exploration of few-shot synthetic data generation, we undertake a comprehensive evaluation of **16 datasets** across a diverse array of tasks typically encountered in clinical NLP benchmarks (Peng et al., 2019; Fries et al., 2022). Specifically, we consider 2 text classification, 3 relation extraction (RE), 3 natural language inference (NLI), 2 fact verification, 1 sentence similarity (STS), 4 NER, and 1 attribute extraction tasks. Please see Appendix D for detailed dataset descriptions and the statistics of each dataset.

**Baselines.** We compare CLINGEN with **9 baselines** in total, including 6 data augmentation and 3 LLM-based data generation techniques. The data augmentation models include Word Substitution (Ribeiro et al., 2020), Back Translation (Xie et al., 2020), Mixup (Chen et al., 2020; Zhang et al., 2020), Transformer (Kumar et al., 2020; Zhou et al., 2022), LightNER (Chen et al., 2022a), and KGPC (Chen et al., 2023a). For LLM-based generation models, we consider ZeroGen (Ye et al., 2022a), DemoGen (Meng et al., 2023; Yoo et al., 2021) and ProGen (Ye et al., 2022b) as representative methods. See Appendix E for details.

### 5.2 MODEL PERFORMANCE WITH THE SYNTHETIC DATA

Table 1 summarizes the experimental results on different datasets. We also conduct supervised learning on the original training data and the extracted few-shot examples, denoted as "Supervised-Full" and "Supervised-Few", respectively. Due to space limits, we report the average performance over all datasets for each task, but provide the detailed results for each dataset in Tables 6, 7, 8 in Appendix G. Based on the experimental results, we have the following findings:

⋄ Our proposed approach, CLINGEN, consistently outperforms the baselines across all tasks. The average performance gain over all *main* metrics is 8.98% at `Base` scale and 7.27% at `Large` scale. In addition, methods utilizing LLMs have better performance than traditional data augmentation techniques, illustrating the capacity of LLMs to extract valuable information from limited examples.

| Task | Single-Sentence Tasks | | Sentence-Pair Tasks | | | | Token Classification Tasks | | | | |
|---|---|---|---|---|---|---|---|---|---|---|---|
| | Text Class | RE | NLI | Fact Verification | | STS | NER | | MedAttr | | |
| | F1 | F1 | Acc | Acc | F1 | Acc | F1 | F1-subset* | P | R | F1 |
| **PubMedBERT_Base** | | | | | | | | | | | |
| Supervised-Full | 77.01 | 77.34 | 79.20 | 67.58 | 65.49 | 75.70 | 89.67 | 87.27 | — | — | — |
| Supervised-Few | 18.61 | 43.89 | 44.64 | 29.43 | 27.10 | 55.70 | 39.41 | 34.12 | 38.11 | 43.82 | 40.77 |
| DA-Word Sub | 40.74 | 38.14 | 55.08 | 28.86 | 25.83 | 54.40 | 44.30 | 40.41 | 40.25 | 47.65 | 43.64 |
| DA-Back Trans | 47.24 | — | 54.30 | 32.15 | 28.04 | 55.80 | — | — | — | — | — |
| DA-Mixup | 45.09 | 43.37 | 53.52 | 32.78 | 29.12 | 58.20 | 42.20 | 37.65 | 42.37 | 48.96 | 45.43 |
| DA-Transformer | 41.02 | 47.56 | 55.71 | 35.32 | 31.77 | 58.80 | 44.75 | 39.66 | 37.82 | 44.28 | 40.80 |
| LightNER† | — | — | — | — | — | — | — | 39.49 | — | — | — |
| KGPC† | — | — | — | — | — | — | — | 51.60 | — | — | — |
| ZeroGen | 59.02 | 63.84 | 55.96 | 35.30 | 32.50 | 68.35 | 56.97 | 48.26 | 52.80 | 49.53 | 51.11 |
| DemoGen | 64.09 | 67.46 | 59.80 | 40.30 | 35.95 | 70.85 | 60.16 | 53.91 | 58.15 | 56.84 | 57.49 |
| ProGen | 65.16 | 67.23 | 59.57 | 37.71 | 34.54 | 69.30 | 60.49 | 55.11 | 57.76 | 58.57 | 58.16 |
| CLINGEN w/ KG | 67.15 | 69.01 | 64.89 | 43.83 | 39.43 | 72.17 | **64.26** | **60.11** | **71.75** | 65.20 | **68.32** |
| CLINGEN w/ LLM | **67.82** | **70.06** | **67.24** | **46.50** | **41.46** | **73.31** | 63.17 | 58.49 | 68.19 | **66.79** | 67.48 |
| Performance Gain | 4.08% | 3.85% | 12.44% | 15.38% | 15.33% | 3.47% | 6.23% | — | — | — | 17.47% |
| **PubMedBERT_Large** | | | | | | | | | | | |
| Supervised-Full | 80.06 | 79.64 | 82.65 | 72.97 | 69.23 | 78.80 | 90.15 | 87.68 | — | — | — |
| Supervised-Few | 17.86 | 52.68 | 50.00 | 40.90 | 30.50 | 59.73 | 42.84 | 37.57 | 41.30 | 45.02 | 43.08 |
| DA-Word Sub | 43.99 | 44.35 | 57.66 | 35.51 | 31.95 | 55.30 | 46.67 | 43.70 | 46.77 | 43.52 | 45.09 |
| DA-Back Trans | 50.98 | — | 58.39 | 34.12 | 31.36 | 56.40 | — | — | — | — | — |
| DA-Mixup | 46.74 | 50.97 | 57.35 | 34.01 | 31.10 | 58.50 | 46.69 | 43.01 | 41.25 | 52.09 | 46.04 |
| DA-Transformer | 44.41 | 46.12 | 58.94 | 35.09 | 30.95 | 58.10 | 46.94 | 43.50 | 43.36 | 45.78 | 44.54 |
| LightNER† | — | — | — | — | — | — | — | — | — | — | — |
| KGPC† | — | — | — | — | — | — | — | — | — | — | — |
| ZeroGen | 61.51 | 65.18 | 63.47 | 41.12 | 36.10 | 72.69 | 57.79 | 49.10 | 54.04 | 51.40 | 52.69 |
| DemoGen | 64.97 | 68.65 | 64.58 | 42.61 | 38.69 | 74.37 | 61.43 | 55.61 | 62.67 | 61.02 | 61.83 |
| ProGen | 65.01 | 69.23 | 63.32 | 42.79 | 38.63 | 74.89 | 62.47 | 57.31 | 57.21 | 63.70 | 60.28 |
| CLINGEN w/ KG | 66.76 | 71.47 | **70.90** | 48.62 | 42.45 | 75.82 | **65.48** | **62.23** | 70.96 | **69.66** | **70.30** |
| CLINGEN w/ LLM | **67.61** | **72.81** | 70.50 | **49.51** | **43.72** | **76.21** | 65.36 | 61.89 | **71.61** | 66.86 | 69.15 |
| Performance Gain | 4.00% | 5.17% | 9.79% | 15.70% | 13.00% | 3.47% | 1.76% | — | — | — | 13.70% |

Table 1: Experimental results aggregated by tasks. **Bold** and underline indicate the best and second best results for each dataset, respectively. †: The models can only be applied to NER tasks, and the number is reported from the original paper. ∗: Since the two † models only report results on two NER datasets, we report an average performance on those two datasets for a fair comparison.

The performance gain of DemoGen and ProGen over ZeroGen further demonstrates the positive influence of few-shot examples on overall comprehension.

⋄ In *token classification tasks*, CLINGEN performs better with KG compared to LLM. This improvement stems from the strong alignment between the task's target and the generated domain knowledge, where the extracted topics serve as direct labels for these datasets. The *single-sentence* and *sentence-pair tasks*, on the other hand, display an advantage for the LLM-based knowledge extraction. This can be attributed to two potential reasons: first, these tasks prioritize understanding entire sentences over specific terminologies, and some specialized terms might even impede LLM comprehension. Second, KGs may not always contain the required information. For example, in a RE task involving chemicals and proteins, some types of the relations are absent from the KG, thus the performance gain is rather limited.

⋄ Some data augmentation methods are task-specific, limiting their generalizability. For example, LightNER and KGPC are designed specifically for NER tasks. It is also non-trivial to apply Back Translation to NER or RE tasks, as it requires locating related entities in the generated sentence accurately. In contrast, CLINGEN is flexible and can be effectively applied to various tasks.

## 5.3  ABLATION AND PARAMETER STUDIES

**Effect of Different LLM Generators.** To investigate the impact of various LLMs on CLINGEN, we leverage other models in the GPT-family as the text generator. Specifically, we utilize Instruct-GPT (`text-curie-001`) (Ouyang et al., 2022) and GPT-4 (OpenAI, 2023b). Note that we only generate 500 samples in the GPT-4 setting due to budget constraints, but we provide the results of GPT-3.5 with same amount of synthetic samples for a fair comparison. From Figure 4 we observe that CLINGEN generally outperforms the best baseline in all settings. Additionally, we observe gen-

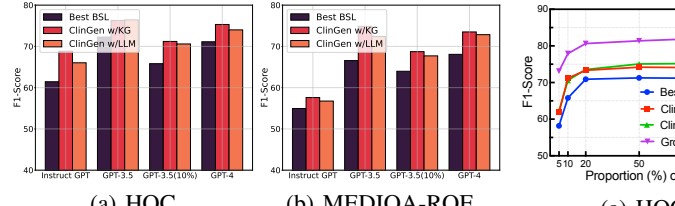

| (a) HOC | (b) MEDIQA-RQE |
|---|---|

Figure 4: Different generators at `Base`.

| (a) HOC | (b) MEDIQA-RQE |
|---|---|

Figure 5: Different proportion of data at `Base`.

| | HOC | GAD | | | ChemProt | MEDIQA-RQE | PUBHEALTH | | NCBI-Disease | | | CASI | | |
|---|---|---|---|---|---|---|---|---|---|---|---|---|---|---|
| | F1 | P | R | F1 | F1 | ACC | ACC | F1 | P | R | F1 | P | R | F1 |
| ChatGPT Inference (OpenAI, 2023a) | 68.76 | 84.21 | **97.46** | 90.35 | 49.42 | 74.31 | **69.50** | **52.47** | 46.62 | 52.31 | 49.30 | 48.82 | **74.75** | 59.07 |
| PMC-LLaMa-7B Inference (Wu et al., 2023) | 32.94 | 90.14 | 90.59 | 90.37 | 13.35 | 52.17 | 14.53 | 2.94 | 61.87 | 37.81 | 46.79 | 59.89 | 37.94 | 45.45 |
| MedAlpaca Inference (Han et al., 2023) | 36.44 | 69.95 | 70.29 | 70.12 | 26.29 | 57.67 | 56.51 | 35.71 | 44.69 | 31.16 | 27.85 | 52.51 | 49.16 | 51.64 |
| CLINGEN w/ KG | 77.71 | 94.30 | 89.09 | **91.62** | 60.12 | **79.92** | 50.20 | 41.26 | **62.46** | **64.08** | **63.26** | 70.96 | 69.66 | **70.30** |
| CLINGEN w/ LLM | **78.14** | **95.08** | 86.14 | 90.39 | **63.05** | 77.36 | 52.96 | 43.31 | 61.12 | 60.16 | 60.64 | **71.61** | 66.86 | 69.15 |

Table 2: Comparison between prompting ChatGPT for inference and CLINGEN at `Large` scale.

erally improved performance with larger models, as they often have better capabilities to follow our designed instructions for the given prompts. See Appendix H for more figures.

**Effect of Size of Synthetic Data.** In Figure 5 (and more in Appendix H), we study the effect of the size of synthetic data. The result shows that CLINGEN consistently outperforms the best baseline, using only around 10% of the synthetic examples. This illustrates that incorporating domain knowledge and increasing the diversity of the prompts could be an effective way to improve the sample efficiency, and narrow the gap between the performance of synthetic and ground-truth dataset.

**Comparison with few-shot inference via prompting ChatGPT.** We also evaluate the performance of few-shot in-context learning with ChatGPT and two medical LLMs, namely PMC-LLaMa (Wu et al., 2023) and MedAlpaca (Han et al., 2023). Due to budget limits, we only run experiments on datasets with few testing samples for each task. As presented in Table 2, CLINGEN at PubMedBERT$_{\texttt{Large}}$ scale achieves better results on 5 out of 6 datasets than ChatGPT few-shot learning, which uses $\sim 530\times$ more parameters. One exception is for PUBHEALTH, as it requires complex reasoning abilities that PubMedBERT$_{\texttt{Large}}$ may not fully possess. The two medical LLMs, on the other hand, perform less effectively than both ClinGen and GPT-3.5 due to fewer parameters, limited reasoning capabilities, and training on a general medical corpus unsuited for the tasks. Overall, CLINGEN offers cost-effective and time-efficient advantages. While it entails a one-time investment in both money and time for synthetic training data generation, subsequent prediction relying on a moderate-sized model is much more efficient. Besides, the continued use of ChatGPT for inference on new testing data incurs ongoing time and financial costs, while our model requires zero additional costs for querying APIs. The price information is exhibited in Appendix J.

**Effect of Topic Extraction and Style Suggestion.** We inspect different components of CLINGEN in Table 3. It is observed that both Topics Extraction and Style Suggestion contribute to model performance as they enhance the relevance of generated samples to domain knowledge and introduce greater diversity. Different from the other datasets, MEDIQA-RQE shows more performance gain incorporating writing style than topics. It is because NLI tasks focus on capturing the relationships between two sentences while incorporating additional knowledge entities does not directly help the model improve the reasoning ability.

| | HOC | | CDR | | MEDIQA-RQE | | NCBI-Disease | |
|---|---|---|---|---|---|---|---|---|
| | w/ KG | w/ LLM | w/ KG | w/ LLM | w/ KG | w/ LLM | w/ KG | w/ LLM |
| CLINGEN | 76.28 | **76.42** | 61.74 | **63.34** | **74.85** | 72.40 | **59.46** | 55.95 |
| w/o Styles | 73.25 | 74.40 | 59.10 | 60.15 | 67.21 | 66.50 | 57.97 | 54.70 |
| w/o Topics | 70.86 | | 58.51 | | 69.86 | | 55.09 | |

Table 3: Ablation studies on topic extraction and style suggestion at `Base` scale.

| | HOC | CDR | MEDIQA-RQE | NCBI-Disease |
|---|---|---|---|---|
| ZeroGen | 0.512 | 0.469 | 0.277 | 0.528 |
| DemoGen | 0.463 | 0.377 | 0.289 | 0.281 |
| ProGen | 0.481 | 0.321 | 0.290 | 0.357 |
| CLINGEN w/ KG | 0.440 | **0.291** | **0.243** | 0.180 |
| CLINGEN w/ LLM | **0.432** | 0.338 | 0.255 | **0.155** |
| Ground truth | 0.265 | 0.268 | 0.164 | 0.262 |

Table 4: Average Pairwise Similarity.

# 6 QUALITY ANALYSIS OF THE SYNTHETIC DATA

**Data Distribution Measures.** In this section, we present the data distribution and diversity measurement of the synthetic dataset generated by CLINGEN. Figure 6(a) shows the t-SNE plot of data

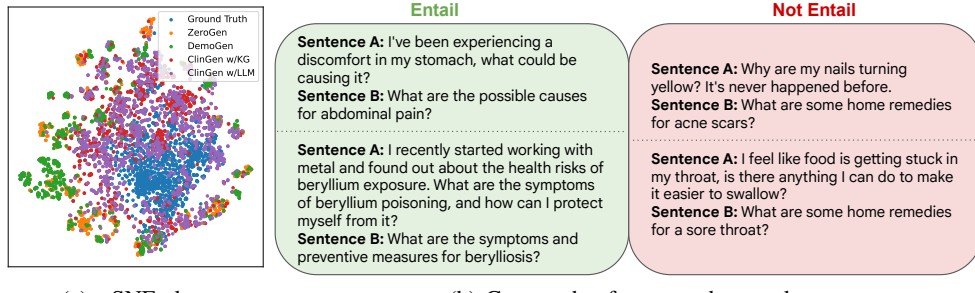

(a) t-SNE plot                         (b) Case study of generated examples

Figure 6: Data distribution and diversity measures on CLINGEN. (a) is from BC5CDR-Disease and (b) is from MEDIQA-RQE using CLINGEN with LLM.

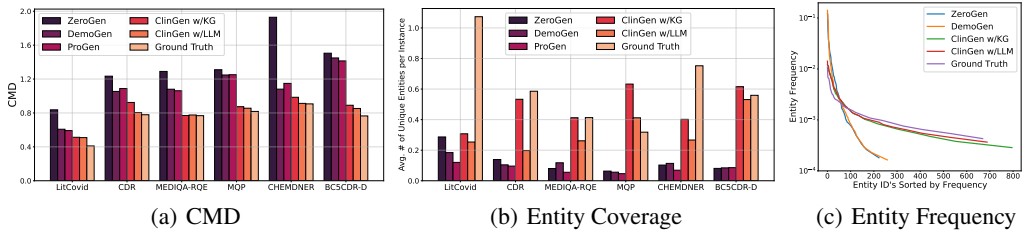

(a) CMD                         (b) Entity Coverage                         (c) Entity Frequency

Figure 7: Data distribution and diversity measures on CLINGEN. (c) is from BC5CDR-Disease.

generated by CLINGEN and baselines compared with the ground truth. This visualization clearly demonstrates that CLINGEN exhibits a greater overlap with the ground truth, indicating a similar distribution as the original dataset. In addition, as depicted in Figure 7(a), the embedding of CLINGEN aligns more closely with the ground truth distribution than other baselines across all six datasets, further justifying the efficacy of CLINGEN for mitigating the distribution shift issue.

**Diversity Measures.** Table 4 calculates the average cosine similarity for sample pairs using SentenceBERT embeddings. Compared to baselines, the dataset generated with CLINGEN exhibits lower cosine similarity and the average similarity is close to that of the ground truth training data, which shows CLINGEN could render more diverse data. Moreover, Figure 7(b) highlights the ability of CLINGEN to cover a broader range of entities in comparison to the baselines. We find that CLINGEN w/ KG captures a larger variety of entities than CLINGEN w/ LLM, because KG tends to cover more extensive knowledge, including relatively uncommon information that may not be present in LLMs. Figure 7(c) reflects that the entity frequency distribution of CLINGEN is more in line with the ground truth, having a relatively balanced distribution among all entities. This ensures that CLINGEN generates synthetic data with a wide range of diverse topics.

**Case Study.** In Figure 6(b), we present a case study of examples generated by CLINGEN with LLM on MEDIQA-RQE dataset, which consists of consumer health queries. The showcased examples reveal that the sentences generated by CLINGEN include more extensive contextual information compared with the baseline as shown in Figure 1(b). These sentences closely resemble the queries people might pose in real-life scenarios.

## 7    CONCLUSION

In this work, we propose a versatile approach to clinical text data generation using LLMs. We thoroughly assess existing methods for clinical data generation and identify issues including distribution shifts and limited diversity. To tackle these challenges, we introduce CLINGEN, a new framework that leverages clinical knowledge from non-parametric KGs and parametric LLMs. This knowledge empowers data generation by utilizing clinical topic knowledge and real-world writing styles in domain-specific prompts. Our extensive empirical evaluations across 7 clinical NLP tasks and 16 datasets, comparing to 9 baseline methods, consistently show that CLINGEN improves task performance, aligns closely with real data, and enhances data diversity. We expect this approach can be seamlessly incorporated into a broad suite of clinical text tasks to advance clinical NLP research.

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

## A    LIMITATION, FUTURE WORKS AND ETHICS ISSUES

In this work, we propose CLINGEN to better harness the LLM for synthetic text data generation. Despite the strong performance of CLINGEN on 16 clinical NLP tasks, we mainly verify their efficacy from their empirical performance, sample diversity, and distribution gaps. However, there still exist gaps between the performance of the model $\mathcal{C}_\theta$ fine-tuned using our generated synthetic data and ground-truth data. To further improve CLINGEN, there are several avenues of future works:

**Using Clinical LLMs as Data Generator**: Our method CLINGEN relies on an LLM with instruction following ability. We mainly evaluate CLINGEN using GPT-family models as the LLM. Recently, there are many LLMs that have been fine-tuned on additional clinical contexts as well as instructions (e.g. Med-PALM[3]), and achieved superior performance on challenging clinical NLP benchmarks. However, they are not open-sourced, thus we cannot run them in our experiments. An interesting future work could be how to leverage these Clinical LLMs as Data Generator to further boost the performance. Besides, it can be beneficial to inject more fine-grained clinical knowledge beyond entity and relations to further benefit data generation pipelines.

**Data Evaluation**: In this work, we consider the distribution gap and sample diversity as our optimization objective. However, there might be many other aspects for synthetic quality estimation (Alaa et al., 2022). We need more tools to capture, analyze, and improve this new aspect of data-centric AI.

**Factuality**: One issue with LLM-based synthetic data generation is the phenomenon of *hallucination*, wherein the model generates information that do not grounded in reality (Zhang et al., 2023). This can lead to the propagation of misinformation, which may have negative impacts on the clinical domain. It is crucial to cross-verify the generated text with a reliable knowledge base or dataset. Furthermore, incorporating an additional layer of human review can also help in mitigating hallucinations and ensuring the faithfulness of LLM-generated synthetic outputs (Zhou et al., 2023a).

**Application to Structured EHR datasets**: On the other hand, EHR data falls within a distinct modality (i.e. tabular data) from textual data, which may require different methodologies and approaches (Ive et al., 2020; Wornow et al., 2023). Nonetheless, we are aware of the capabilities of LLMs in this context. Recent studies (Hegselmann et al., 2023; Borisov et al., 2022) have explored transforming tabular data into text to harness the power of LLMs, which yields promising results and shows the potential of LLMs for structured data generation. However, as these approaches are fundamentally different from the methods we propose in this paper, they are beyond the scope of this paper.

**Privacy concerns:** We are well aware of the patient privacy concern in clinical NLP. Specifically, we carefully curate the five few-shot demonstrations to ensure they only contain conceptual information and are fully free from any Protected Health Information (PHI) related to patients. With the five de-identified examples as the only data input for demonstrations, the synthetic training data we generate is highly *unlikely to include any private information*. We also acknowledge the possibility of inadvertently introducing sensitive data through the GPT model itself. To address this, we make a deliberate effort to avoid any instructions that can potentially extract sensitive patient information within the prompts. Instead, the prompts we use focus solely on obtaining conceptual information relevant to the target task. Lastly, we conduct rigorous inspections of the generated synthetic data across all covered tasks to affirm that no such private information exists in the synthetic data generated by our method.

## B    ADDITIONAL PRELIMINARY STUDIES

We present additional preliminary studies of the t-SNE plots in Figure 8 and the regularized entity frequencies in Figure 9. These results further justify the distribution shift issue mentioned in section 3.2, demonstrating that the limited diversity as well as the distribution shift issue generally exists for a broad range of clinical NLP tasks.

---

[3]https://sites.research.google/med-palm/

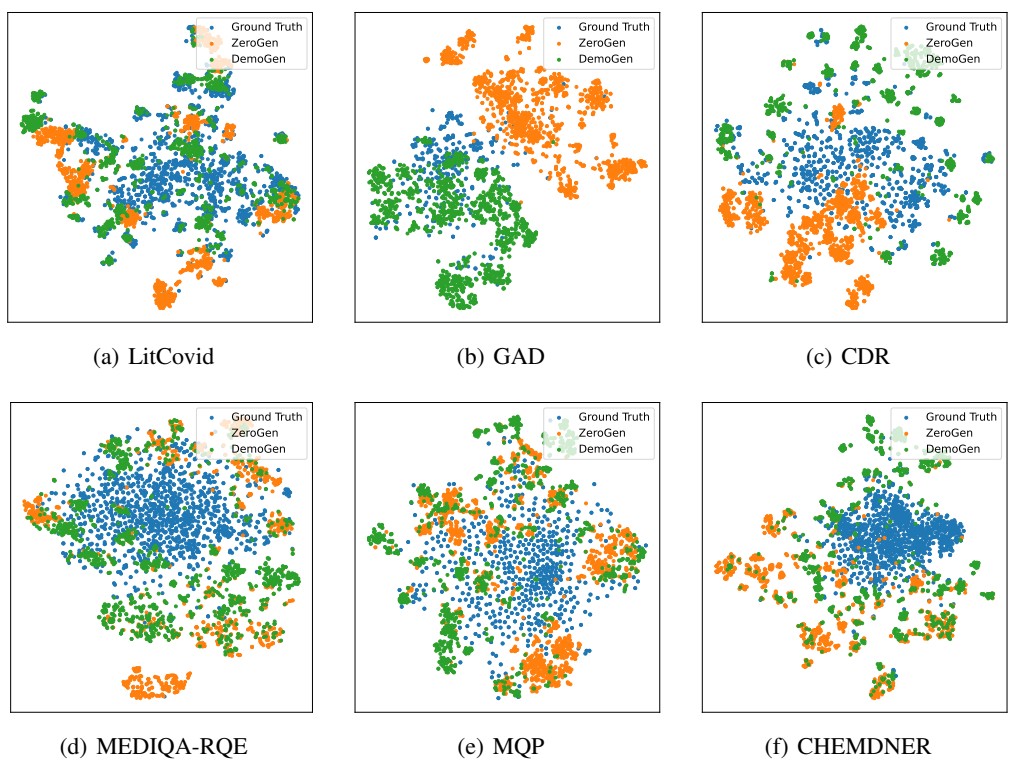

(a) LitCovid     (b) GAD     (c) CDR

(d) MEDIQA-RQE     (e) MQP     (f) CHEMDNER

Figure 8: The t-SNE plots of datasets generated by ZeroGen and DemoGen compared with the ground truth.

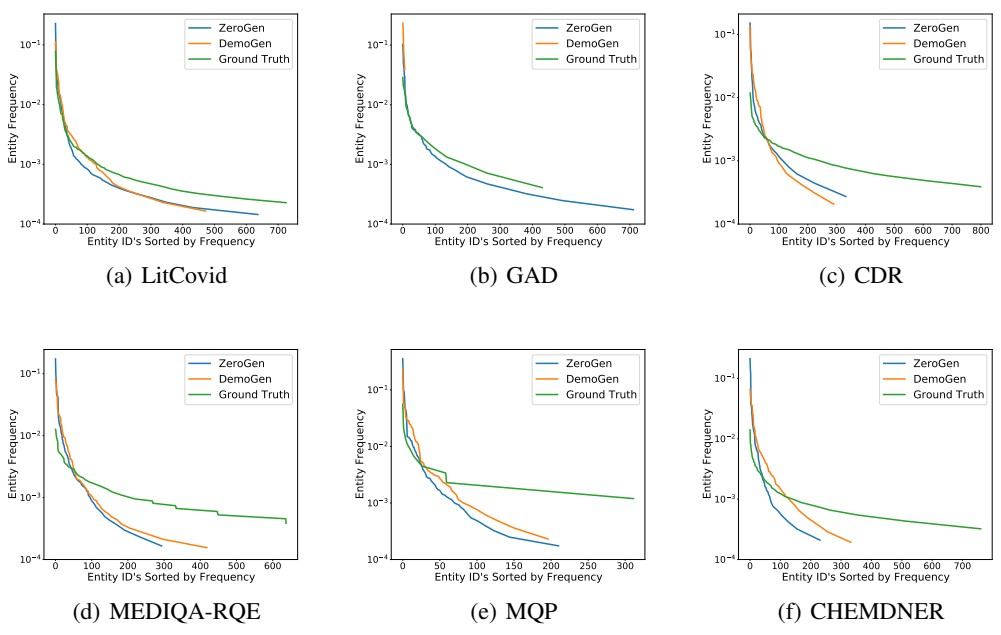

(a) LitCovid     (b) GAD     (c) CDR

(d) MEDIQA-RQE     (e) MQP     (f) CHEMDNER

Figure 9: The regularized entity frequencies of datasets generated by ZeroGen and DemoGen compared with the ground truth in log scale.

## C    IMPLEMENTATION DETAILS

For implementation, we use PyTorch (Paszke et al., 2019) and HuggingFace (Wolf et al., 2019). For each dataset, we randomly sample 5 examples from each class to provide few-shot demonstrations and keep a validation set of the same size. In the experiments, We generate 5000 synthetic training data for both CLINGEN and the baselines and report the average performance over 3 random seeds for all the results.

During the data generation process when we call the ChatGPT APIs (OpenAI, 2023a), we set the parameter $\text{top\_p} = 1.0$ and temperature $t = 1.0$ to balance between the quality of the generated text as well as diversity (Chung et al., 2023; Yu et al., 2023)[4]. With the generated synthetic dataset, we follow the common few-shot learning setting (Perez et al., 2021) to train all the models for 6 epochs and use the model with the best performance on the validation set for evaluation.

During the PubMedBERT fine-tuning, we adopt AdamW (Loshchilov & Hutter, 2017) for optimization with a linear warmup of the first 5% steps and linear learning rate decay. The learning rate is set to 2e-5 for `Base` and 4e-5 for `Large`, and the maximum number of tokens per sequence is 256.

## D    DATASET DESCRIPTION

| Corpus | Tasks | #Class | #Train/#Test | Metrics |
|---|---|---|---|---|
| **Single-Sentence Tasks** | | | | |
| LitCovid (Chen et al., 2021) | Text Classification | 7 | 24960/6238 | **F1** |
| HOC (Baker et al., 2015) | Text Classification | 10 | 3091/898 | **F1** |
| GAD (Bravo et al., 2015) | Relation Extraction (RE) | 1 | 4750/350 | P, R, **F1** |
| CDR (Wei et al., 2016) | Relation Extraction (RE) | 1 | 8431/2522 | P, R, **F1** |
| ChemProt (Taboureau et al., 2010) | Relation Extraction (RE) | 5 | 8793/1087 | **F1** |
| **Sentence-Pair Tasks** | | | | |
| MedNLI* (Shivade, 2017) | Natural Language Inference (NLI) | 3 | 11232/1422 | **Acc** |
| MEDIQA-NLI† (Ben Abacha et al., 2019) | Natural Language Inference (NLI) | 3 | -/405 | **Acc** |
| MEDIQA-RQE (Abacha & Demner-Fushman, 2016) | Natural Language Inference (NLI) | 2 | 8588/302 | **Acc** |
| PUBHEALTH (Kotonya & Toni, 2020) | Fact Verification | 4 | 9804/1231 | Acc, **F1** |
| HealthVer (Sarrouti et al., 2021) | Fact Verification | 3 | 10591/1824 | Acc, **F1** |
| MQP (McCreery et al., 2020) | Sentences Similarity (STS) | 2 | 10/3033 | **Acc** |
| **Token Classification Tasks** | | | | |
| BC5CDR-Disease (Li et al., 2016a) | Named Entity Recognition (NER) | 1 | 4882/5085 | P, R, **F1** |
| BC5CDR-Chemical (Li et al., 2016a) | Named Entity Recognition (NER) | 1 | 4882/5085 | P, R, **F1** |
| NCBI-Disease (Dogan et al., 2014) | Named Entity Recognition (NER) | 1 | 5336/921 | P, R, **F1** |
| CHEMDNER (Krallinger et al., 2015) | Named Entity Recognition (NER) | 1 | 14522/12430 | P, R, **F1** |
| CASI (Agrawal et al., 2022; Moon et al., 2014) | Attribute Extraction | 6 | 5/100 | **F1** |

Table 5: Dataset statistics. We do not count the non-entity/non-relation class for relation extraction and token classification tasks to align with existing works. P and R stand for Precision and Recall. Metrics in **bold** are considered as the main metrics. * is not allowed to put into GPT and † does not provide training data, so we sample few-shot examples from the SciTail (Khot et al., 2018) instead.

The evaluation tasks and datasets are summarized in Table 5. Note that the number of training samples indicates the size of the *original* training set. Specifically, we consider the following datasets:

- **Single-Sentence Tasks**
  - Text Classification:
    - * The *LitCovid* dataset (Chen et al., 2021) consists of COVID-19-related publications from PubMed. The task is to predict the topics of the sentences, including "Epidemic Forecasting", "Treatment", "Prevention", "Mechanism", "Case Report", "Transmission", and "Diagnosis".
    - * The *HOC* dataset (Baker et al., 2015) also extracts sentences from PubMed articles, each annotated at the sentence level. The task is to predict the topics of the sentences, including "evading growth suppressors", "tumor promoting inflammation", "enabling replicative

---

[4]We do not further increase $t$, as previous analysis (Chung et al., 2023; Yu et al., 2023) has shown that increasing $t$ to larger value does not help with additional performance gain.

immortality", "cellular energetics", "resisting cell death", "activating invasion and metastasis", genomic instability and mutation", "inducing angiogenesis", "sustaining proliferative signaling", and "avoiding immune destruction".

- ○ Relation Extraction:
  - ∗ The *GAD* (Bravo et al., 2015) dataset is to predict whether there is a relation between the given disease and gene in the sentences. Note that the original annotation for this dataset is Noisy. To remedy this issue, we *relabel* 350 examples from the original test set to form a clean subset for faithful evaluation.
  - ∗ The *CDR* (Wei et al., 2016) dataset is to predict whether the provided chemical can induce the disease in the sentences.
  - ∗ The *ChemProt* (Taboureau et al., 2010) dataset focuses on the chemical-protein relations, and the labels include "Upregulator", "Downregulator", "Agonist", "Antagonist", "Product_of" and "No relation".

- **Sentence-Pair Tasks**
  - ○ Natural Language Inference (NLI):
    - ∗ The *MedNLI* (Shivade, 2017) dateset consists of sentences pairs derived from MIMIC-III, where we predict the relations between the sentences. The labels include "entailment", "neutral" and "contradiction".
    - ∗ The *MEDIQA-NLI* (Ben Abacha et al., 2019) dataset comprises text-hypothesis pairs. Their relations include "entailment", "neutral" and "contradiction".
    - ∗ The *MEDIQA-RQE* (Abacha & Demner-Fushman, 2016) dataset contains NIH consumer health question pairs, and the task is to recognize if the first question can entail the second one.
  - ○ Fact Verification:
    - ∗ The *PUBHEALTH* (Kotonya & Toni, 2020) encompasses claims paired with journalist-crafted explanations. The task is to predict the relations between the claim and evidence, including "Refute", "Unproven", "Support", and "Mixture".
    - ∗ The *HealthVer* (Sarrouti et al., 2021) contains evidence-claim pairs from search engine snippets regarding COVID-19 questions. The relations between claims and evidences are chosen from "Refute", "Unproven", and "Support".
  - ○ Sentence Similarity (STS):
    - ∗ the *MQP* (McCreery et al., 2020) dataset comprises a collection of medical question pairs designed for identifying semantically similar questions. The task is to predict whether the two questions are equivalent or not.

- **Token Classification Tasks**
  - ○ Named Entity Recognition (NER):
    - ∗ The *BC5CDR-Disease* (Li et al., 2016b) is to recognize diseases in the sentences.
    - ∗ The *BC5CDR-Chemical* (Li et al., 2016b) is to recognize chemicals in the sentences.
    - ∗ The *NCBI-Disease* (Dogan et al., 2014) is to recognize diseases in the sentences.
    - ∗ The *CHEMDNER* (Krallinger et al., 2015) is to recognize chemicals in the sentences.
  - ○ Attribute Extraction (MedAttr):
    - ∗ The *CASI* dataset (Agrawal et al., 2022; Moon et al., 2014) aims to identify interventions including medication, dosage, route, freq, reason, duration

## E  BASELINE DETAILS

In this section, we give a detailed description for all baselines used in this study.
**Data Augmentation Methods:**

- **DA-Word Sub**: It performs word substitution for few-shot demonstrations to create new training sample. Specifically, we follow Checklist (Ribeiro et al., 2020) and maintain a word list to generate new examples.

- **DA-Back Translation**: It employ back translation to augment the training data Xie et al. (2020), including translating text from the target language to the source language and then back to the target language.

- **DA-Mixup** (Chen et al., 2020; Zhang et al., 2020): It adds interpolation on the *embedding space* of the training examples to create virtual augmented examples. In this work, we use the TMix version of MixText for data augmentation on the few-shot labeled dataset. For token-level classification tasks, we employ one variant of Mixup, namely SeqMix (Zhang et al., 2020) that leverage the interpolation on the embedding space to generate new tokens as augmentations.

- **DA-Transformer (MELM)** (Kumar et al., 2020; Zhou et al., 2022): It introduces a conditional data augmentation technique that prepends class labels to text sequences for pretrained transformer-based models. Specifically, it leverage the sequence to sequence transformer (BART) to perform conditional text generation based on the seed examples. For token-level classification tasks, we select MELM (Zhou et al., 2022), which first masks some entities in the few-shot examples, and use a text-to-text transformer (T5) to predict masked entity tokens by explicitly conditioning on their labels for creating new examples.

- **LightNER** (Chen et al., 2022a): It adopts a seq2seq framework, generating the entity span sequence and entity categories under the guidance of a self-attention-based prompting module. It is designed specifically for NER tasks.

- **KGPC** (Chen et al., 2023a): It injects the semantic relations of the knowledge graph to sequence to sequence text generation models to perform knowledge-guided instance generation for few-shot biomedical NER. It also only applies to NER tasks.

**LLM-based Generation Methods.**

- **ZeroGen** (Ye et al., 2022a): It generates a dataset using simple class-conditional prompts and then trains a tiny task-specific model for zero-shot inference. We follow the prompting method mentioned in their original paper as implementation, which *does not consider* any style information as well as domain knowledge.

- **DemoGen** (Meng et al., 2023; Yoo et al., 2021): It leverages LLMs to synthesize novel training data by feeding few-shot samples as demonstrations to guide the data generation process without providing additional instructions. Note that we focus on using the black-box LLM as the generator, thus we do not tune the LLM as Meng et al. (2023).

- **ProGen** (Ye et al., 2022b): It leverages the feedback from the task-specific model to guide the generation of new training data via in-context examples. Specifically, it first identify the most important examples from the generated synthetic data using the influence function, then added these examples as demonstrations to generate new training instances. To ensure fair comparison, we also add the few-shot demonstrations for data generation.

We do not compare with Tang et al. (2023) in the main experiments as it leverages entities extracted from the entire training set and violates the true few-shot learning setting.

## F  PROMPT FORMAT

### F.1  THE PROMPTS FOR WRITING STYLES SUGGESTION WITH CLINGEN

Listing 1: Prompt Format for writing styles suggestion with CLINGEN.

```
Suppose you need to generate a synthetic clinical text dataset on
[task] tasks. Here are a few examples from the original training
set:
[demonstrations]
Please write three potential sources, speakers or authors of the
sentences.
```

`[task]`: The task names for each specific task. `[demonstrations]`: The few-shot demonstrations from the original training set.

### F.2  THE PROMPTS FOR DATA GENERATION WITH CLINGEN

In the following prompt format, `[topic]` and `[style]` are randomly sampled from the topics candidate set and styles candidate set we formulate in the knowledge extraction step, respectively.

**Named entity recognition tasks:**

Listing 2: Prompt Format for NER tasks with CLINGEN.

```
Suppose you need to create a dataset for [domain] recognition.
Your task is to:
1. generate a sentence about [domain],
2. output a list of named entity about [domain] only,
3. the sentence should mimic the style of [style],
4. the sentence should mention the [domain] named [topic].
```

[domain]: "disease" for BC5CDR-Disease and NCBI-Disease; "chemical" for BC5CDR-Chemical and CHEMDNER.

**Medication attributes tasks:**

Listing 3: Prompt Format for medication attributes tasks with CLINGEN.

```
Suppose you need to create a dataset for clinical attributes
recognition. Your task is to:
1. generate a sentence about clinical attributes, The Clinical
Attributes you need to extract include "Medication", "Dosage", "
Route", "Frequency", "Reason", "Duration". For each attribute
class, please return a list of attributes within the class that
occurs in the Sentence.
2. the sentence should mimic the style of [style],
3. the sentence should be relevant to [topic].
```

**Text classification tasks:**

Listing 4: Prompt Format for text classification tasks with CLINGEN.

```
 Suppose you need to create a dataset for [domain]. Your task is
 to:
 1. generate a sentence about [domain].
 2. the sentence should mimic the style of [style].
 3. the sentence should be relevant to the subtopic of [topic] for
  [class_name].
```

[domain]: "COVID-19 Literature" for LitCovid and "Cancer Document" for HOC.

[class_name]: the label name for this generated sample.

**Relation extraction tasks:**

Listing 5: Prompt Format for relation extraction tasks with CLINGEN.

```
Suppose you need to generate synthetic data for the biomedical
[domain] task. Your task is to:
1. give a sentence about [class_name] relation between [entity0]
and [entity1]
2. the sentence should discuss the [entity0]: [topic0] and
[entity1]: [topic1] with the relation [label_desc].
3. the sentence should mimic the style of [style].
```

[domain]: "Disease Gene Relation" for GAD, "Chemical Disease Relation" for CDR, and "Chemical Protein Relation" for ChemProt.

[entity0] and [entity1]: "disease" and "gene" for GAD, "chemical" and "disease: for CDR, and "chemical" and "protein" for ChemProt.

[class_name]: the label name for this generated sample.

[label_desc]: the description of the selected label. For example, the label "upregulator" in ChemProt has a description of "the chemical activates expression of the protein."

**Natural language inference tasks:**

Listing 6: Prompt Format for generating the first sentence in NLI tasks with CLINGEN.

```
Suppose you need to create a set of [content]. Your task is to:
1. generate one sentence for a [content].
2. the [content] should be relevant to [topic],
3. The [content] should mimic the style of [style].
```

[content]: "health question" for MEDIQA-RQE, "claim" for MEDIQA-NLI, MedNLI and MQP, and "health news" for PUBHEALTH and HealthVer.

Listing 7: Prompt Format for generating the second sentence in NLI tasks with CLINGEN.

```
Suppose you need to create a pair of sentences for the [domain]
task with the label '[class_name]'. Given the [content]: '
[first_sentence]', Your task is to:
1. generate one short [content] about [topic] so that
[label_desc].
2. The [content] should mimic the style of the first sentence.
```

[domain]: "Question Entailment" for MEDIQA-RQE, "Natural Language Entailment" for MEDIQA-NLI and MedNLI, "Fact Verification" for PUBHEALTH and HealthVer, and "Sentence Similarity Calculation" for MQP.

[content]: "health question" for MEDIQA-RQE, "hypothesis" for MEDIQA-NLI, MedNLI, "evidence" for PUBHEALTH and HealthVer, and "sentence" for MQP.

[class_name]: the label name for this generated sample.

[label_desc]: the description of the selected label.

[first_sentence]: the first sentence we generate

### F.3 PROMPTS FOR ZEROGEN, DEMOGEN, PROGEN

We use the same set of prompts for ZeroGen, DemoGen and ProGen, while DemoGen and ProGen have additional demonstrations augmented to the prompts. DemoGen uses the few-shot examples in the training set as demonstrations, and ProGen leverages feedbacks from previous rounds to iteratively guide the generation.

**Named entity recognition tasks:**

Listing 8: Prompt Format for NER tasks with baselines.

```
Suppose you need to create a dataset for [domain] recognition.
Your task is to generate a sentence about [domain] and output a
list of named entity about [domain] only.
```

[domain]: "disease" for BC5CDR-Disease and NCBI-Disease; "chemical" for BC5CDR-Chemical and CHEMDNER.

**Medication attributes tasks:**

Listing 9: Prompt Format for medication attributes tasks with baselines.

```
Suppose you need to create a dataset for clinical attributes
recognition. Your task is to generate a sentence about clinical
attributes, The Clinical Attributes you need to extract include "
Medication", "Dosage", "Route", "Frequency", "Reason", "Duration".
 For each attribute class, please return a list of attributes
within the class that occurs in the Sentence.
```

**Text classification tasks:**

Listing 10: Prompt Format for text classification tasks with baselines.

```
Suppose you are a writer for [domain]. Your task is to give a
synthetic [domain] about [class_name].
```

[domain]: "COVID-19 Literature" for LitCovid and "Cancer Document" for HOC.

[class_name]: the label name for this generated sample.

**Relation extraction tasks:**

Listing 11: Prompt Format for relation extraction tasks with baselines.

```
Suppose you need to generate synthetic data for the biomedical
[domain] task. Your task is to give a sentence about [class_name]
relation between [entity0] and [entity1] so that [label_desc].
```

[domain]: "Disease Gene Relation" for GAD, "Chemical Disease Relation" for CDR, and "Chemical Protein Relation" for ChemProt.

[entity0] and [entity1]: "disease" and "gene" for GAD, "chemical" and "disease: for CDR, and "chemical" and "protein" for ChemProt.

[class_name]: the label name for this generated sample.

[label_desc]: the description of the selected label. For example, the label "upregulator" in ChemProt has a description of "the chemical activates expression of the protein."

**Natural language inference tasks:**

Listing 12: Prompt Format for generating the first sentence in NLI tasks with baselines.

```
Suppose you need to create a set of [content]. Your task is to
generate one sentence for a [content].
```

[content]: "health question" for MEDIQA-RQE, "claim" for MEDIQA-NLI, MedNLI and MQP, and "health news" for PUBHEALTH and HealthVer.

Listing 13: Prompt Format for generating the second sentence in NLI tasks with baselines.

```
Suppose you need to create a pair of sentences for the [domain]
task with the label '[class_name]'. Given the [content]: '
[first_sentence]', Your task is to generate one short [content] so
 that [label_desc].
```

[domain]: "Question Entailment" for MEDIQA-RQE, "Natural Language Entailment" for MEDIQA-NLI and MedNLI, "Fact Verification" for PUBHEALTH and HealthVer, and "Sentence Similarity Calculation" for MQP.

[content]: "health question" for MEDIQA-RQE, "hypothesis" for MEDIQA-NLI, MedNLI, "evidence" for PUBHEALTH and HealthVer, and "sentence" for MQP.

[class_name]: the label name for this generated sample.

[label_desc]: the description of the selected label.

[first_sentence]: the first sentence we generate

## G   ADDITIONAL EXPERIMENTAL RESULTS

In this section, we present additional experimental results on every dataset in Tables 6, 7, 8. We also include the experimental results combining topic from both KG and LLM, which yields a performance improvement, though not a substantial one. However, note that in practice, it is challenging to tune the ratio in the few-shot setting.

| | LitCovid | HOC | CDR | | | GAD | | | ChemProt |
|---|---|---|---|---|---|---|---|---|---|
| | F1 | F1 | P | R | F1 | P | R | F1 | F1 |
| **PubMedBERT_Base** | | | | | | | | | |
| Supervised-Full (SOTA) | 73.55 | 84.35 | 67.81 | 76.60 | 71.96 | — | — | 84.39 | 77.97 |
| Supervised-Full | 71.70 | 82.32 | 67.81 | 76.60 | 71.96 | 82.55 | 85.10 | 83.81 | 76.24 |
| Supervised-Few | 24.08 | 13.13 | 41.62 | 52.96 | 46.61 | 57.71 | 46.54 | 51.53 | 33.54 |
| DA-Word Sub | 36.49 | 44.98 | 40.50 | 46.20 | 43.16 | 51.15 | 32.10 | 39.45 | 31.82 |
| DA-Back Trans | 39.7 | 54.78 | — | — | — | — | — | — | — |
| DA-Mixup | 40.82 | 49.35 | 41.4 | 44.8 | 43.03 | 55.44 | 48.30 | 51.62 | 35.45 |
| DA-Transformer | 39.86 | 42.18 | 44.6 | 61.7 | 51.77 | 59.4 | 46.5 | 52.16 | 38.73 |
| ZeroGen | 50.50 | 67.90 | 38.82 | **91.82** | 54.57 | 84.38 | 80.68 | 82.49 | 54.46 |
| DemoGen | 57.65 | 70.52 | 46.9 | 83.3 | 60.01 | 93.14 | 80.19 | 86.18 | 56.18 |
| ProGen | 58.06 | 72.25 | 51.35 | 71.58 | 59.80 | 90.52 | **85.14** | 87.75 | 54.15 |
| CLINGEN w/ KG | 58.01 | 76.28 | 56.98 | 67.38 | 61.75 | 93.33 | 83.68 | **88.24** | 57.04 |
| CLINGEN w/ LLM | **59.22** | **76.42** | **60.6** | 66.35 | **63.34** | **94.61** | 78.17 | 85.61 | **61.22** |
| CLINGEN w/ KG+LLM | 56.56 | 78.02 | 57.97 | 71.09 | 63.86 | 92.57 | 88.59 | 90.54 | 58.48 |
| **PubMedBERT_Large** | | | | | | | | | |
| Supervised-Full (SOTA) | — | 84.87 | — | — | — | — | — | 84.90 | 78.77 |
| Supervised-Full | 74.59 | 85.53 | 72.31 | 74.88 | 73.57 | 84.95 | 88.75 | 86.81 | 78.55 |
| Supervised-Few | 22.59 | 13.13 | 42.27 | 67.51 | 51.99 | 57.58 | 90.07 | 70.25 | 35.80 |
| DA-Word Sub | 37.20 | 50.78 | 47.70 | 43.50 | 45.50 | 63.40 | 42.00 | 50.53 | 37.01 |
| DA-Back Trans | 40.50 | 61.46 | — | — | — | — | — | — | — |
| DA-Mixup | 40.03 | 53.45 | 43.34 | 73.50 | 54.53 | 62.20 | 59.93 | 60.52 | 37.87 |
| DA-Transformer | 38.95 | 49.86 | 50.70 | 31.60 | 38.93 | 59.80 | 57.76 | 58.76 | 40.66 |
| ZeroGen | 52.86 | 70.16 | 42.95 | **80.67** | 56.06 | 92.26 | 76.73 | 83.78 | 55.71 |
| DemoGen | 56.29 | 73.65 | 50.86 | 74.30 | 60.39 | **96.85** | 76.83 | 85.69 | 59.88 |
| ProGen | 54.71 | 75.31 | 50.36 | 76.08 | 60.60 | 91.11 | 85.63 | 88.29 | 58.79 |
| CLINGEN w/ KG | 55.81 | 77.71 | 60.45 | 65.04 | 62.66 | 94.30 | **89.08** | **91.62** | 60.12 |
| CLINGEN w/ LLM | **57.07** | **78.14** | **67.13** | 62.98 | **64.99** | 95.08 | 86.14 | 90.39 | **63.05** |
| CLINGEN w/ KG+LLM | 56.80 | 79.07 | 64.19 | 67.70 | 65.90 | 92.41 | 92.07 | 92.24 | 59.95 |

Table 6: Performance on single-sentence tasks evaluated by PubMedBERT_Base and PubMedBERT_Large. **Bold** and underline indicate the best and second best results for each dataset, respectively. Note that the performance of 'Supervised-Full (SOTA)' is copied from the existing paper. If the value in this field is missing, this means we cannot find reported results with the same-scale model on that dataset. (Same as below).

| | MEDIQA-RQE | MEDIQA-NLI | MedNLI | PUBHEALTH | | HealthVer | | MQP |
|---|---|---|---|---|---|---|---|---|
| | ACC | ACC | ACC | ACC | F1 | ACC | F1 | ACC |
| **PubMedBERT_Base** | | | | | | | | |
| Supervised-Full (SOTA) | — | — | 86.60 | 70.52 | 69.73 | 73.54 | 74.82 | 79.20 |
| Supervised-Full | 77.15 | 79.01 | 81.43 | 65.16 | 62.96 | 70.00 | 68.02 | 75.70 |
| Supervised-Few | 57.51 | 40 | 36.40 | 28.30 | 23.70 | 30.55 | 30.49 | 55.70 |
| DA-Word Sub | 58.60 | 50.24 | 56.4 | 23.67 | 17.64 | 34.05 | 34.02 | 54.40 |
| DA-Back Trans | 59.16 | 49.92 | 53.82 | 30.70 | 23.32 | 33.60 | 32.76 | 55.80 |
| DA-Mixup | 57.71 | 49.38 | 53.47 | 31.45 | 24.45 | 34.11 | 33.78 | 58.20 |
| ZeroGen | 63.28 | 52.89 | 57.71 | 35.80 | 31.50 | 34.80 | 33.50 | 68.35 |
| DemoGen | 66.56 | 56.29 | 58.56 | 42.60 | 35.40 | 38.00 | 36.50 | 70.85 |
| ProGen | 65.94 | 57.28 | 59.49 | 38.70 | 33.10 | 36.72 | 35.97 | 69.30 |
| CLINGEN w/ KG | **74.85** | 58.03 | 61.80 | 44.60 | 36.80 | 43.05 | 42.06 | 72.17 |
| CLINGEN w/ LLM | 72.40 | **64.44** | **64.89** | **48.50** | **40.60** | **44.50** | **42.32** | **73.31** |
| CLINGEN w/ KG+LLM | 75.10 | 64.12 | 65.81 | 50.57 | 40.65 | 40.60 | 39.59 | 68.30 |
| **PubMedBERT_Large** | | | | | | | | |
| Supervised-Full (SOTA) | — | — | 86.57 | — | — | — | — | 81.00 |
| Supervised-Full | 81.10 | 82.89 | 83.96 | 70.21 | 63.45 | 75.72 | 75.01 | 78.80 |
| Supervised-Few | 63.79 | 47.40 | 38.80 | 46.20 | 27.20 | 35.60 | 33.80 | 59.73 |
| DA-Word Sub | 64.26 | 51.20 | 57.53 | 35.60 | 31.60 | 35.41 | 32.29 | 55.30 |
| DA-Back Trans | 65.52 | 51.43 | 58.21 | 34.45 | 30.50 | 33.78 | 32.21 | 56.40 |
| DA-Mixup | 64.10 | 50.91 | 57.03 | 34.23 | 30.78 | 33.79 | 31.42 | 58.50 |
| ZeroGen | 67.26 | 60.74 | 62.42 | 42.50 | 33.30 | 39.74 | 38.90 | 72.69 |
| DemoGen | 69.22 | 62.97 | 64.55 | 44.50 | 36.80 | 40.72 | 40.57 | 74.37 |
| ProGen | 67.82 | 60.98 | 63.15 | 44.15 | 36.37 | 41.42 | 40.89 | 74.89 |
| CLINGEN w/ KG | **79.92** | 63.59 | 69.19 | 50.20 | 41.26 | **47.03** | 43.64 | 75.82 |
| CLINGEN w/ LLM | 77.36 | **64.69** | **69.46** | **52.96** | **43.31** | 46.05 | **44.12** | **76.21** |
| CLINGEN w/ KG+LLM | 80.77 | 63.30 | 70.56 | 51.98 | 41.61 | 47.44 | 44.25 | 71.90 |

Table 7: Performance on sentence-pair tasks evaluated by PubMedBERT_Base and PubMedBERT_Large.

| | BC5CDR-Disease | | | BC5CDR-Chemical | | | NCBI-Disease | | | CHEMDNER | | | CASI | | |
|---|---|---|---|---|---|---|---|---|---|---|---|---|---|---|---|
| | P | R | F1 | P | R | F1 | P | R | F1 | P | R | F1 | P | R | F1 |
| **PubMedBERT_Base** | | | | | | | | | | | | | | | |
| Supervised-Full (SOTA) | — | — | 86.10 | — | — | 93.33 | — | — | 88.76 | — | — | 92.35 | — | — | — |
| Supervised-Full | 83.84 | 87.92 | 85.83 | 92.22 | 91.74 | 91.98 | 87.54 | 89.92 | 88.71 | 91.84 | 92.45 | 92.14 | — | — | — |
| Supervised-Few | 24.86 | 39.47 | 30.51 | 63.73 | 46.07 | 53.48 | 36.16 | 39.47 | 37.74 | 48.00 | 28.70 | 35.92 | 38.11 | 43.82 | 40.77 |
| DA-Word Sub | 35.34 | 39.54 | 37.32 | 63.13 | 52.52 | 57.34 | 53.40 | 36.70 | 43.50 | 47.45 | 33.15 | 39.03 | 40.25 | 47.65 | 43.64 |
| DA-Mixup | 36.13 | 42.90 | 39.23 | 66.43 | 50.54 | 57.41 | 56.57 | 26.48 | 36.07 | 42.37 | 48.96 | 45.43 | | | |
| LightNER | 39.80 | 33.20 | 36.20 | — | — | — | 43.70 | 41.90 | 42.78 | — | — | — | | | |
| DA-MELM | 34.20 | 41.30 | 37.42 | 47.23 | 72.81 | 57.29 | 36.90 | 48.50 | 41.91 | 39.33 | 45.95 | 42.38 | 37.82 | 44.28 | 40.80 |
| KGPC | 50.80 | 51.30 | 51.05 | — | — | — | 52.20 | 52.10 | 52.15 | — | — | — | | | |
| ZeroGen | 55.60 | 39.10 | 45.91 | 73.20 | 82.85 | 77.73 | 56.25 | 45.98 | 50.60 | 54.34 | 52.93 | 53.63 | 52.80 | 49.53 | 51.11 |
| DemoGen | 63.10 | 48.44 | 54.81 | 76.40 | 81.65 | 78.94 | 57.65 | 49.08 | 53.02 | 54.00 | 53.77 | 53.88 | 58.15 | 56.84 | 57.49 |
| ProGen | 61.60 | 50.5 | 55.50 | 77.10 | 82.02 | 79.48 | 56.01 | 53.50 | 54.73 | 51.55 | 53.00 | 52.26 | 57.76 | 58.57 | 58.16 |
| CLINGEN w/ KG | 58.64 | 63.02 | 60.75 | 74.96 | 85.45 | 79.86 | 62.62 | 56.62 | 59.47 | 48.33 | 69.28 | 56.94 | 71.75 | 65.20 | 68.32 |
| CLINGEN w/ LLM | 63.41 | 58.83 | 61.03 | 77.68 | 84.33 | 80.87 | 62.58 | 50.59 | 55.95 | 51.40 | 58.77 | 54.84 | 68.19 | 66.79 | 67.48 |
| CLINGEN w/ KG+LLM | 60.57 | 66.21 | 63.26 | 73.66 | 87.30 | 79.90 | 58.01 | 65.37 | 59.17 | 52.07 | 63.62 | 57.27 | 72.57 | 70.48 | 71.51 |
| **PubMedBERT_Large** | | | | | | | | | | | | | | | |
| Supervised-Full (SOTA) | — | — | 86.39 | — | — | 94.04 | — | — | 89.18 | — | — | 92.72 | — | — | — |
| Supervised-Full | 86.77 | 85.92 | 86.34 | 92.80 | 92.94 | 92.87 | 87.97 | 90.09 | 89.02 | 92.23 | 92.48 | 92.35 | — | — | — |
| Supervised-Few | 25.52 | 45.85 | 32.79 | 61.40 | 54.41 | 57.69 | 44.86 | 40.12 | 42.35 | 43.40 | 34.60 | 38.50 | 41.30 | 45.02 | 43.08 |
| DA-Word Sub | 38.54 | 38.85 | 38.69 | 64.85 | 53.96 | 58.91 | 52.59 | 45.35 | 48.70 | 44.85 | 36.69 | 40.36 | 46.77 | 43.52 | 45.09 |
| DA-Mixup | 36.27 | 46.67 | 40.82 | 67.63 | 54.15 | 60.14 | 55.64 | 38.06 | 45.20 | 45.51 | 36.66 | 40.61 | 41.25 | 52.09 | 46.04 |
| LightNER | — | — | — | — | — | — | — | — | — | — | — | — | — | — | — |
| DA-MELM | 33.40 | 41.61 | 37.06 | 53.80 | 66.71 | 59.56 | 44.20 | 57.40 | 49.94 | 36.40 | 47.41 | 41.18 | 43.36 | 45.78 | 44.54 |
| KGPC | — | — | — | — | — | — | — | — | — | — | — | — | — | — | — |
| ZeroGen | 57.40 | 39.21 | 46.59 | 78.08 | 80.97 | 79.49 | 54.52 | 49.00 | 51.61 | 48.56 | 59.44 | 53.45 | 54.04 | 51.40 | 52.69 |
| DemoGen | 57.34 | 49.48 | 53.12 | 78.27 | 83.90 | 80.99 | 59.43 | 56.83 | 58.10 | 48.03 | 60.39 | 53.51 | 62.67 | 61.02 | 61.83 |
| ProGen | 60.34 | 54.13 | 57.07 | 78.42 | 82.94 | 80.62 | 60.02 | 55.28 | 57.55 | 50.40 | 59.64 | 54.63 | 57.21 | 63.70 | 60.28 |
| CLINGEN w/ KG | 54.28 | 70.14 | 61.21 | 77.88 | 86.32 | 81.88 | 62.46 | 64.08 | 63.26 | 47.03 | 67.86 | 55.56 | 70.96 | 69.66 | 70.30 |
| CLINGEN w/ LLM | 61.05 | 65.40 | 63.15 | 78.08 | 86.98 | 82.29 | 61.12 | 60.16 | 60.64 | 50.92 | 60.67 | 55.37 | 71.61 | 66.86 | 69.15 |
| CLINGEN w/ KG+LLM | 65.67 | 66.22 | 65.94 | 75.89 | 87.61 | 81.33 | 65.70 | 59.22 | 62.31 | 52.49 | 65.07 | 58.11 | 73.21 | 69.30 | 71.20 |

Table 8: Performance on token-classification tasks evaluated by PubMedBERT_Base and PubMedBERT_Large.

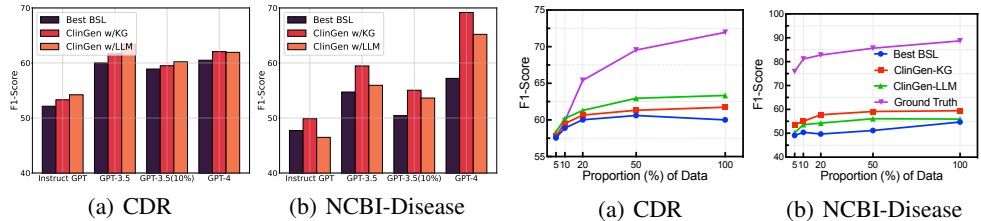

| (a) CDR | (b) NCBI-Disease | | (a) CDR | (b) NCBI-Disease |
|---|---|---|---|---|

Figure 10: Different generators at `Base`.  Figure 11: Different proportion of data at `Base`.

|   | HOC | | | CDR | | | MEDIQA-RQE | | | NCBI-Disease | | |
|---|---|---|---|---|---|---|---|---|---|---|---|---|
|   | Best Baseline | CLINGEN-KG | CLINGEN-LLM | Best Baseline | CLINGEN-KG | CLINGEN-LLM | Best Baseline | CLINGEN-KG | CLINGEN-LLM | Best Baseline | CLINGEN-KG | CLINGEN-LLM |
| 1 | 70.04 | 74.30 | 77.30 | 61.52 | 61.66 | 63.34 | 68.30 | 76.85 | 74.50 | 56.12 | 60.22 | 54.51 |
| 2 | 75.30 | 79.73 | 73.63 | 60.69 | 63.77 | 64.66 | 64.20 | 71.80 | 71.19 | 54.19 | 60.64 | 57.81 |
| 3 | 71.41 | 74.81 | 78.33 | 57.82 | 59.79 | 62.02 | 67.18 | 75.90 | 71.51 | 53.85 | 57.52 | 55.50 |

Table 9: Performance with Different Random Seeds using PubMedBERT$_{\text{Base}}$.

# H ADDITIONAL ABLATION AND PARAMETER STUDIES

Figure 10 and 11 show the effect of different generators and the effect of the proportion of data on two additional datasets, respectively. Overall, our method generally outperform the best baseline. One interesting finding for the NCBI-Disease dataset is that CLINGEN performs worse than the best on one variant. We hypothesize that it is because this task involves more complex input and output, potentially posing a challenge for moderate-size LLMs to follow the instructions.

Besides, as few-shot sample selection is important for the final performance, we show the performance of different 3 random seeds (with different seed examples/training process), and observe that our method CLINGEN generally outperforms the baselines with non-negligible margins, which indicates the robustness of CLINGEN as it does not rely on a specific subset of few-shot training examples to perform well.

# I ADDITIONAL QUALITY ANALYSIS

We present additional quality analysis of the synthetic dataset with t-SNE plots in Figure 12 and the regularized entity frequencies in Figure 13.

# J MONETARY COST

We display the monetary cost of CLINGEN for calling the OpenAI APIs, with a comparison with prompting GPT-3.5 for direct inference and DemoGen. From the values shown in Figure 10, we observe that inference via GPT-3.5 generally has a higher cost, as it needs to input all the testing samples for prompting. In contrast, DemoGen has a relatively lower cost, because it does not include the topics and writing styles to the prompts as CLINGEN does.

|   | HOC | GAD | ChemProt | MEDIQA-RQE | PUBHEALTH | NCBI-Disease | CASI |
|---|---|---|---|---|---|---|---|
| GPT-3.5 Inference | 1.09 | 1.05 | 5.75 | 2.15 | 2.80 | 0.90 | 1.30 |
| DemoGen | 0.59 | 0.66 | 1.35 | 0.81 | 0.92 | 1.12 | 1.28 |
| CLINGEN w/ KG | 0.65 | 0.73 | 1.47 | 0.86 | 1.01 | 1.41 | 1.55 |
| CLINGEN w/ LLM | 0.72 | 0.84 | 1.51 | 0.90 | 1.34 | 1.49 | 1.62 |

Table 10: The average cost (in US dollars) of running CLINGEN on various datasets per 1000 samples, compared with prompting GPT-3.5 for inference and DemoGen.

# K ADDITIONAL EXPERIMENTS ON MULTI-CHOICE QA DATASETS

We conduct additional experiments on two QA datasets, namely BioASQ (Tsatsaronis et al., 2015) and PubMedQA (Jin et al., 2019) on ClinGen and the most relevant baselines. As presented in Table 11, ClinGen consistently outperforms the baselines on these two datasets.

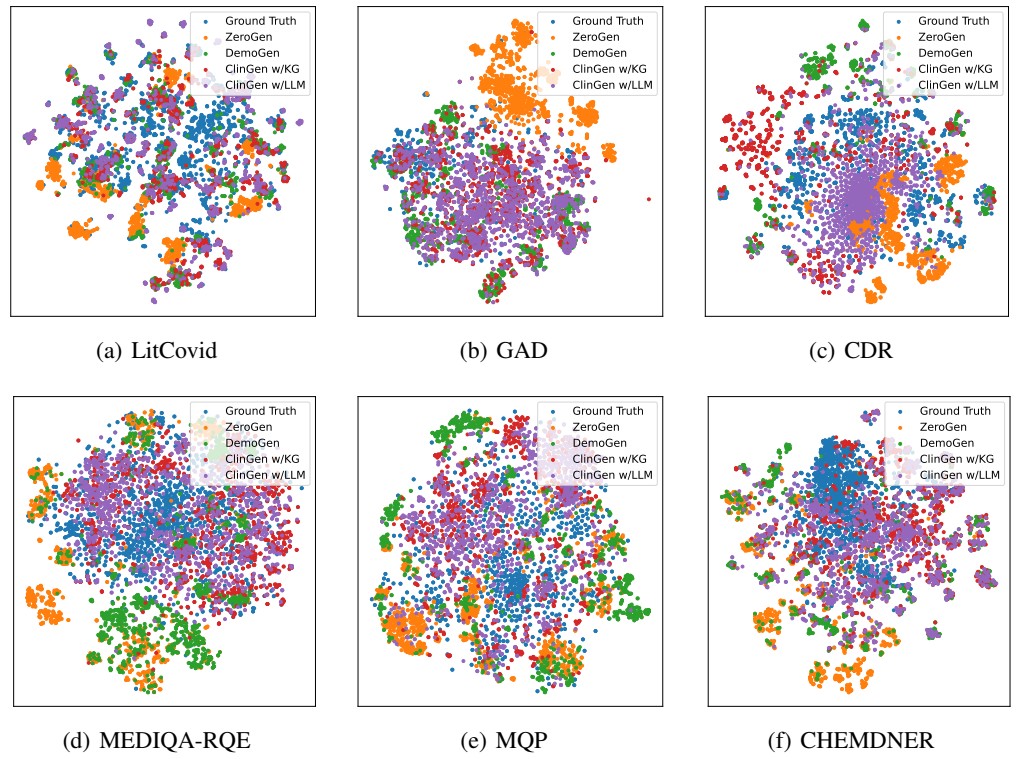

Figure 12: The t-SNE plots of datasets generated by CLINGEN, ZeroGen and DemoGen compared with the ground truth.

However, when it comes to more intricate QA tasks designed to generate free-form answers. It is challenging to establish quantitative metrics that reliably correlate with human accuracy judgments (Chen et al., 2019), even with the assistance of state-of-the-art LLMs (Chen et al., 2023b). Usually, human evaluators are required to assess the answer quality, which might also introduce subjectivity and scalability issues into the evaluation process.

|  | **BioASQ** | **PubMedQA** |
|---|---|---|
|  | ACC | ACC |
| **PubMedBERT_Base** |  |  |
| ZeroGen | 64.57 | 52.68 |
| DemoGen | 62.71 | 54.65 |
| ProGen | 65.71 | 54.83 |
| CLINGEN w/ KG | 66.44 | 56.85 |
| CLINGEN w/ LLM | **67.14** | **57.52** |
| **PubMedBERT_Large** |  |  |
| ZeroGen | 68.66 | 55.05 |
| DemoGen | 68.26 | 55.28 |
| ProGen | 67.42 | 58.87 |
| CLINGEN w/ KG | 69.25 | **61.79** |
| CLINGEN w/ LLM | **70.71** | 61.12 |

Table 11: The performance of CLINGEN and the most relevant baselines on two QA datasets.

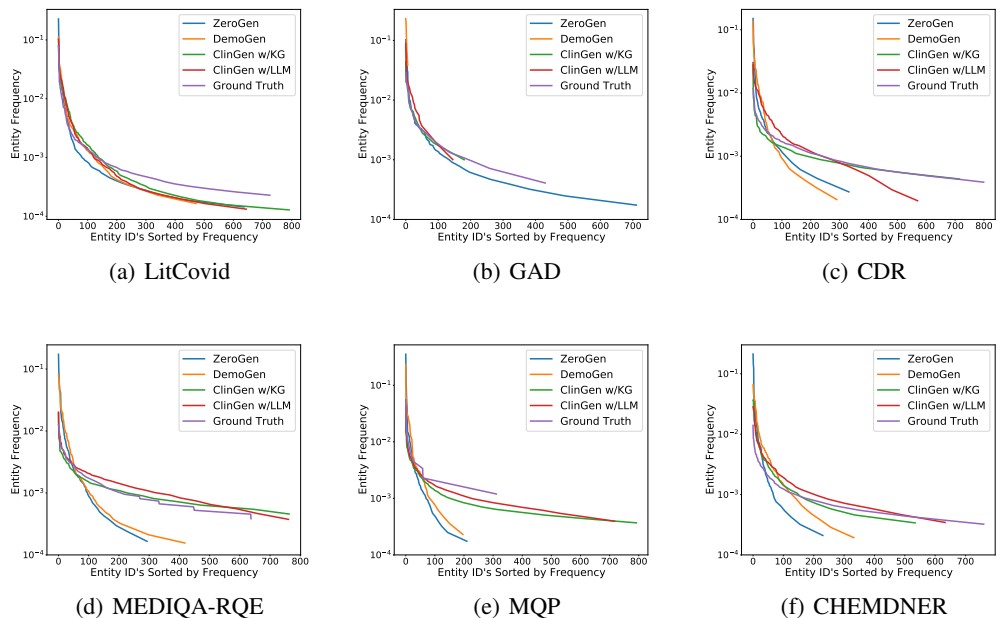

(a) LitCovid        (b) GAD        (c) CDR

(d) MEDIQA-RQE        (e) MQP        (f) CHEMDNER

Figure 13: The regularized entity frequencies of datasets generated by CLINGEN, ZeroGen and DemoGen compared with the ground truth in log scale.

|  | LitCovid | CDR | MEDIQA-RQE | MQP | CHEMDNER | BC5CDR-Disease | Average |
|---|---|---|---|---|---|---|---|
|  | F1 | F1 | ACC | ACC | F1 | F1 | – |
| **PubMedBERT_Base** | | | | | | | |
| Reframe (Mishra et al., 2022) | 56.74 | 57.27 | 61.92 | 67.60 | 54.61 | 59.17 | 59.55 |
| APE (Zhou et al., 2023b) | 56.24 | 61.12 | 66.55 | 68.00 | 52.10 | 58.79 | 60.47 |
| CLINGEN W/ KG | 58.01 | 61.75 | **74.85** | 72.17 | **56.94** | 60.75 | **64.08** |
| CLINGEN W/ LLM | **59.22** | **63.34** | 72.40 | **73.31** | 54.84 | **61.03** | 64.02 |
| **PubMedBERT_Large** | | | | | | | |
| Reframe (Mishra et al., 2022) | 54.06 | 58.78 | 66.57 | 71.30 | 55.05 | 60.41 | 61.03 |
| APE (Zhou et al., 2023b) | 53.54 | 61.65 | 69.20 | 71.00 | 53.03 | 59.87 | 61.38 |
| CLINGEN W/ KG | 55.81 | 62.66 | **79.92** | 75.82 | **55.56** | 61.21 | 65.16 |
| CLINGEN W/ LLM | **57.07** | **64.99** | 77.36 | **76.21** | 55.37 | **63.15** | **65.69** |

Table 12: Comparison between existing prompting optimization methods and CLINGEN.

## L  COMPARISON WITH DIFFERENT PROMPT DESIGNS

### L.1  MODEL PERFORMANCE

we carry out an additional analysis with two recent and representative prompt optimization techniques, namely Reframe (Mishra et al., 2022) and APE (Zhou et al., 2023b). In our setting, Reframe incorporates several principles (e.g. using low-level patterns, itemizing instructions, etc.) to produce high-quality prompts to enhance text generation, whereas APE leverages the LLM itself to automatically optimize the prompts based on the target task information. We demonstrate their performance on various clinical tasks in Table 12.

The results indicate that our proposed CLINGEN consistently outperforms both baselines. This performance gain is attributed to the fact that the prompts generated by Reframe and APE mainly focus on incorporating and decomposing task-specific information, but do not *adequately address* the unique challenges for the clinical data generation task, i.e. distribution shift and lack of diversity.

### L.2  PROMPT TEMPLATES

We provide the detailed prompt templates we use for Reframe (Mishra et al., 2022) and APE (Zhou et al., 2023b) in the followings.

**Natural Language Inference tasks:**

Listing 14: Prompt Format for generating sentences in NLI tasks with Reframe.

```
Generate a pair of sentences for the [domain] task. Follow these
guidelines:
1. Formulate a medical premise in the first sentence, such as a
clinical observation or a patient's medical history.
2. Craft a medical hypothesis or claim related to the premise in
the second sentence.
3. Ensure that the hypothesis logically follows from the premise.
4. Avoid introducing any unrelated or contradictory information in
 either sentence.
5. The length should be in 50 words.
```

Listing 15: Prompt Format for generating sentences in NLI tasks with APE.

```
Generate a pair of sentences for the [domain] task. The first
sentence should be a medical premise, such as a clinical
observation or a patient's medical history. The second sentence
should be a medical hypothesis or claim, related to the premise.
The goal is to determine whether the hypothesis logically follows
from the premise, and you can use various medical scenarios,
conditions, or treatments for creating these sentence pairs.
```

[domain]: "Question Entailment" for MEDIQA-RQE.

**Sentence similarity tasks:**

Listing 16: Prompt Format for generating sentences in sentence similarity tasks with Reframe.

```
Suppose you need to generate two sentences for the [domain] task.
Your task is to give a pair of sentences with the following
instructions:
(1) Generate two sentences that exhibit a clear similarity or
dissimilarity in meaning without using complex or specialized
terms.
(2) express attributes affirmatively.
(3) Ensure that both sentences have a common attribute for
comparison.
(4) The length should be in 50 words.
```

Listing 17: Prompt Format for generating sentences in sentence similarity tasks with APE.

```
Suppose you need to generate two sentences for the [domain] task.
The goal is to assess how close or similar the meaning of two
sentences is, including 'equivalent' or 'not equivalent'.
```

[domain]: "Sentence Similarity Calculation" for MQP.

**Text classification tasks:**

Listing 18: Prompt Format for generating sentences in text classification tasks with Reframe.

```
Suppose you are a writer for [domain]. Your task is to give a
synthetic [domain] about [class_name] with the following
instructions:
(1) Illustrate points with everyday scenarios related to the
[class_name].
(2) about 50 - 100 words.
```

Listing 19: Prompt Format for generating sentences in text classification tasks with APE.

```
Suppose you are a writer for [domain]. Generate a clinical article
 discussing the latest advancements in [domain] with a focus on
[class_name]. Please include information on recent clinical trials
, emerging research findings, and potential implications for
healthcare practitioners and patients.
```

[domain]: "COVID-19 Literature" for LitCovid.

[class_name]: the label name for this generated sample.

**Relation extraction tasks:**

Listing 20: Prompt Format for generating sentences in relation extraction tasks with Reframe.

```
Suppose you need to generate a dataset for the biomedical
[domain] task where the relationships between entities in
biomedical texts need to be identified. Your task is to give a
synthetic example about [class_name] relation with the following
instructions:
(1) Provide the sentence or text snippet where the relationship is
 mentioned.
(2) The length should be in 50 words.
```

Listing 21: Prompt Format for relation extraction tasks with APE.

```
Generate a sentence that describes a [class_name] [domain] between
 [entity0] and [entity1]. The sentence should provide information
about how these terms are related, such as its potential
therapeutic use, side effects, or any relevant research findings.
```

[domain]: "Chemical Disease Relation" for CDR.

[entity0] and [entity1]: "chemical" and "disease: for CDR.

[class_name]: the label name for this generated sample.

**Named entity recognition tasks:**

Listing 22: Prompt Format for generating sentences in NER tasks with Reframe.

```
Suppose you need to create a dataset for [domain] recognition.
Your task is to generate a sentence about [domain] and also output
 the [domain] name with the following instructions:
(1) Generate a sentence that contains a named entity. The named
entity should be a recognizable entity type within the sentence.
(2) The named entity must be contextually relevant and correctly
labeled with its type.
(3) The length should be in 50 words.
```

Listing 23: Prompt Format for NER tasks with APE.

```
Suppose you need to create a dataset for [domain] recognition.
Generate a sentence or short text passage where you mention a
[domain] entity within a context. The named entity should be
clearly identifiable within the text.
```

[domain]: "disease" for BC5CDR-Disease; "chemical" for CHEMDNER.

