# OpenReview forum: "Knowledge-Infused Prompting: Assessing and Advancing Clinical Text Data Generation with Large Language Models"
_ICLR.cc/2024/Conference — Submitted to ICLR 2024_

### Official Review · Reviewer_zJ5m · 2023-10-13

**Soundness:** 3 good
**Presentation:** 2 fair
**Contribution:** 2 fair
**Rating:** 6
**Confidence:** 4

**Summary:**

The paper described a method to fill in templates with terms taken from medical domain knowledge graphs and LLM suggestions. Once the prompt are obtained they are fed to ChatGPT to create synthetic datasets that can be used for MLM fine-tuning and later used for wide variety of NLP tasks.

**Strengths:**

The concept of using LLMs to generate training data is important and should be explored more to achieve best practices.

**Weaknesses:**

- The paper mentioned privacy as an issue but I think that the method will not address any privacy issues that are accruing in the original LLM
- Hard to understand what is the base model used for MLM training, is it PubMedBERT? If so, PubMedBERT outperforms all of the other attempts
- Baselines descriptions could be more elaborated

**Questions:**

The most important question is what is the additional pre trained classifier
It's unclear from the paper

---

> ### Author Response · Authors · 2023-11-19
> **Initial Response to Reviewer zJ5m**
>
> We deeply thank the reviewer for these detailed suggestions on improving the presentation of our work.
>
> ***
> > W1: The paper mentioned privacy as an issue but I think that the method will not address any privacy issues that are accruing in the original LLM.
>
> A: Thanks for the insightful feedback. We have answered this question in the general response for all reviewers. Please refer to the **Patient privacy concerns** section for details.
>
> ***
> > W2: Hard to understand what is the base model used for MLM training, is it PubMedBERT? If so, PubMedBERT outperforms all of the other attempts.
>
> A: Thanks for your question. We would like to kindly point out that for the base model, “MLM training” is usually conducted during the pretraining stage, while in our work, we focus on the **fine-tuning** stage where we adopt the cross-entropy loss towards the target task with the generated synthetic data (see Section 4.3 for details). This is a common setting that is widely used for evaluating the quality of synthetic text data [1,2,3], where better performance indicates better synthetic data utility.
>
> We note that for the base model, we use the same PubMedBERT for ClinGen and baselines to **ensure fair comparison**. For the PubMedBERT model, we fine-tune it for different target tasks using the synthetic text data **without additional MLM training**. As evidenced in Table 1, our method, leveraging the same base model (PubMedBERT-base and large), consistently outperforms the baselines. This better performance is attributed primarily to the high quality of the generated data.
>
> > [1] Ye et al. "Zerogen: Efficient zero-shot learning via dataset generation." EMNLP 2022.
> >
> > [2] Meng et al. "Tuning language models as training data generators for augmentation-enhanced few-shot learning." ICML 2023.
> >
> > [3] Ye et al. "ProGen: Progressive zero-shot dataset generation via in-context feedback." EMNLP 2022
>
> ***
> > W3: Baselines descriptions could be more elaborated
>
> A: Thank you for the suggestion. We provide the baseline details in **Appendix E** in our original manuscript, and we have expanded these descriptions to provide more comprehensive information in the revision.
>
> ***
> > Q1: The most important question is what is the additional pre trained classifier It's unclear from the paper
>
> A: As mentioned in W2, we utilize the same pre-trained PubMedBERT as a classifier for our method and baselines on downstream tasks to prevent unfair comparison. Please see “Sec 5.1 Experiment Setup” for reference. The word “additional” is to differentiate the classifier with the previous LLM generator. We do not conduct any additional pre-training. We have deleted the word to avoid confusions.
>
> ***
> Thanks again for your review! Please let us know if you have any further questions.

---

### Official Review · Reviewer_CY6f · 2023-11-01

**Soundness:** 3 good
**Presentation:** 4 excellent
**Contribution:** 3 good
**Rating:** 8
**Confidence:** 5

**Summary:**

This paper proposes CLINGEN to generate synthetic clinical text data using LLMs to enhance clinical NLP models. The prompt for LLMs incorporates clinical information and the generated synthetic data has a similar distribution as the original datasets, is more diverse, and helps NLP models perform better in few-shot settings (7 clinical NLP tasks, 16 datasets, compared to 9 baseline methods).

**Strengths:**

1. The work has conducted thorough experiments on 7 clinical NLP tasks and 16 datasets in the few-shot settings and shows a clear advantage of the proposed method CLINGEN. The limitations of previous related works are also clearly discussed, quantified and demonstrated. The proposed method has shown promising results to mitigate the limitations.

2. The work has potential significance in the clinical NLP domain, where gold standard data can be limited and private patient needs to be protected. Synthetic data generation can be useful to address both challenges. The work has shown a promising avenue to utilise synthetic clinical data.

3. The paper is overall well-written and clear.

**Weaknesses:**

1. The prompt design is quite simple, i.e. the method itself lacks novelty.

2. The works claim to "infuse" clinical knowledge into the prompt but the prompt simply incorporates clinical topics/concepts (picked from a knowledge graph and LLMs) like "Stoke", rather than actual clinical knowledge (for example, diabetes, stroke, and CHF are risk factors for CKD). Using the word "infuse clinical knowledge" over-claims the novelty of the method.

3. It is unclear how the prompts in Appendix F are determined. Also, the work does not experiment with different prompt designs.

**Questions:**

One useful and missing reference: Ive J, Viani N, Kam J, Yin L, Verma S, Puntis S, Cardinal RN, Roberts A, Stewart R, Velupillai S. Generation and evaluation of artificial mental health records for natural language processing. NPJ digital medicine. 2020 May 14;3(1):69.

The paper did not discuss patient privacy in clinical NLP. Synthetic data should be anonymous and can help AI researchers build models without touching real private patient data.

Textual data is an important component in clinical data but structured data (vital signs, lab tests) is also critical. This work focuses on clinical text synthetic data generation only and more types of data can be considered as future works. However, the capability of LLMs to generate synthetic structured clinical data is questionable.

The clinical topics/concepts in section 4.1.1 are picked from KG and LLMs. How many topics are from KGs and how many are from LLMs? How is this ratio determined? Would a different ratio impact results? Why not ask a clinician to hand-pick clinical topics?

The current SOTA results (regardless of fully supervised or few-shot) should be added to Table 1 as reference points.

D_train in Equation 1 has only K (K<=5) samples per label?

---

> ### Author Response · Authors · 2023-11-19
> **Initial Response to Reviewer CY6f (Part 1)**
>
> We thank the reviewer for the constructive feedback. We answer your questions as follows.
>
> ***
> > W1: The prompt design is quite simple, i.e. the method itself lacks novelty.
>
> A: Thanks for the insightful feedback. We have answered in the general response for all reviewers. Please refer to the **Lack of novelty** section for details.
>
> ***
> > W2: The works claim to "infuse" clinical knowledge into the prompt but the prompt simply incorporates clinical topics/concepts (picked from a knowledge graph and LLMs) like "Stoke", rather than actual clinical knowledge (for example, diabetes, stroke, and CHF are risk factors for CKD). Using the word "infuse clinical knowledge" over-claims the novelty of the method.
>
> A: Thank you for the valuable suggestion. While we would like to kindly point out that we do include certain relational information for relation extraction tasks, we acknowledge that we did not define the scope of the "clinical knowledge" integrated into our model. Accordingly, we have made revisions in the introduction to explicitly specify that we employ clinical concepts and basic relationships among them as the form of clinical knowledge in our study. We have also recognized the potential for incorporating more fine-grained clinical knowledge as part of our future work, which is detailed in **Appendix A**.
>
> ***
> > W3: It is unclear how the prompts in Appendix F are determined. Also, the work does not experiment with different prompt designs.
>
> A: Thanks for the feedback. We notice that the majority of existing research on synthetic data generation relies on manually crafted, straightforward prompts without intricate designs [1,2,3]. Thus, we design our prompt based on the structures of those existing works and task-specific information mentioned in [4], with additional instructions to further incorporate the selected clinical topics and writing styles. For other existing prompt optimization methods in the literature, we identify two related techniques that may be potentially applicable to our case. Please refer to the **Comparison with different prompt designs** section of the general response to all reviewers for the experimental results and further discussions.
>
> >[1] Ye et al. "Zerogen: Efficient zero-shot learning via dataset generation." EMNLP 2022.
> >
> >[2] Meng et al. "Tuning language models as training data generators for augmentation-enhanced few-shot learning." ICML 2023.
> >
> >[3] Ye et al. "ProGen: Progressive zero-shot dataset generation via in-context feedback." EMNLP 2022.
> >
> >[4] Fries et al. "Bigbio: a framework for data-centric biomedical natural language processing." NeurIPS 2022.
>
> ***
> > Q1: One useful and missing reference: Ive J, Viani N, Kam J, Yin L, Verma S, Puntis S, Cardinal RN, Roberts A, Stewart R, Velupillai S. Generation and evaluation of artificial mental health records for natural language processing. NPJ digital medicine. 2020 May 14;3(1):69.
>
> A: Thanks for mentioning this relevant paper. We have modified our manuscript and added the reference in **Section 1** and **Appendix A**.
>
> ***
> > Q2: The paper did not discuss patient privacy in clinical NLP. Synthetic data should be anonymous and can help AI researchers build models without touching real private patient data.
>
> A: Thanks for the insightful feedback. We have answered this question in the general response for all reviewers. Please refer to the **Patient privacy concerns** section for details.
>
> ***
> > Q3: Textual data is an important component in clinical data but structured data (vital signs, lab tests) is also critical. This work focuses on clinical text synthetic data generation only and more types of data can be considered as future works. However, the capability of LLMs to generate synthetic structured clinical data is questionable.
>
> A: Thanks for the thoughtful feedback. We have answered this question in the general response for all reviewers. Please refer to the **Application to QA and EHR tasks** section for details.

---

> ### Author Response · Authors · 2023-11-19
> **Initial Response to Reviewer CY6f (Part 2)**
>
> > Q4: The clinical topics/concepts in section 4.1.1 are picked from KG and LLMs. How many topics are from KGs and how many are from LLMs? How is this ratio determined? Would a different ratio impact results?
>
> A: Thank you for bringing up this question. In our experiments, we keep the topics from KGs and LLMs **separate**. Specifically, we create a candidate set of approximately 6000 topics from KGs and another candidate set of around 2000 topics from LLMs. We then independently sample topics from these two candidate sets to generate two distinct sets of synthetic training samples. Please refer to **Table 1** for the experimental results for both sets of synthetic data.
>
> The motivation behind this is that, in our paper, we present KGs and LLMs as two alternative and complementary sources for obtaining topics for readers to understand the influence of topic injection from KG and LLMs distinctly. However, we agree with the reviewer that combining topics from KGs and LLMs has the potential to enhance performance. Thus, we have conducted additional experiments to demonstrate the impact of combining topics from KGs and LLMs at various ratios. Note that we still keep a total of 5000 generated synthetic samples to maintain a fair comparison.
>
> | KG : LLM| LitCovid | CDR | MEDIQA-RQE | BC5CDR-D | Average |
> |:--:|:--:|:--:|:--:|:--:|:--:|
> | PubMedBERT-Base | F1 | F1 | ACC | F1 ||
> | 1:0 | 58.01 | 61.75 | 74.85 | 60.75 | 63.84 |
> | 2:1 | 56.18 | 62.89 | 73.50 | 60.53 | 63.28 |
> | 1:1 | 56.76 | 63.86 | 74.01 | 63.26 | 64.47 |
> | 1:2 | 55.49 | 64.33 | 75.10 | 61.62 | 64.14 |
> | 0:1 | 59.22 | 63.34 | 72.40 | 61.03 | 64.00 |
> | PubMedBERT-Large ||||||||
> | 1:0 | 55.81 | 62.66 | 79.92 | 61.21 | 64.90 |
> | 2:1 | 54.21 | 64.22 | 76.15 | 62.40 | 64.25 |
> | 1:1 | 56.80 | 65.90 | 79.12 | 65.94 | 66.94 |
> | 1:2 | 54.41 | 64.68 | 80.77 | 64.55 | 66.10 |
> | 0:1 | 57.07 | 64.99 | 77.36 | 63.15 | 65.64 |
>
> The experimental results indicate that combining knowledge from KGs and LLMs can yield a performance improvement, though not a substantial one. However, note that in practice, it is challenging to tune the ratio in the few-shot setting due to the limited volume of validation labels [5], and thus we only include the 1:1 results in **Table 6,7,8** in **Appendix G** for all 16 datasets.
>
> > [5] Perez et al. "True few-shot learning with language models." NeurIPS 2021.
>
> ***
> > Q4 (continued): Why not ask a clinician to hand-pick clinical topics?
>
> A: Thank you for the suggestion. While we admit clinicians' ability to select clinical topics, it is important to point out that such a manual approach involves significant time and money, thus is less scalable. Furthermore, it can be challenging for clinicians to manually create thousands of clinical topics for synthetic training samples. To this end, we utilize KGs and LLMs as more efficient methods to automatically extract clinical topics.

---

> ### Author Response · Authors · 2023-11-19
> **Initial Response to Reviewer CY6f (Part 3)**
>
> > Q5: The current SOTA results (regardless of fully supervised or few-shot) should be added to Table 1 as reference points.
>
> A: Thanks for the valuable suggestion. In our study, we present the supervised model performance on few-shot demonstrations and the complete original training set, denoted as "Supervised - Few" and "Supervised - Full," respectively. We use PubMedBERT, the same classifier employed for ClinGen and all baseline models, as a reference point.
>
> Following the reviewer's request, we have tried our best to gather relevant data on SOTA models and their performances. However, this task is nontrivial due to several reasons: (1) Many studies only test their models on a subset of the datasets we used; (2) Original papers for some datasets might report results only for base models (like BERT) and not include more recent models (such as PubMedBERT or LinkBERT); (3) In some cases, datasets report only F1 scores without precision and recall metrics.
>
> Despite our best efforts to compile this information, there are still missing values in the available SOTA performance data. We have documented these findings in Tables 6, 7, and 8 in Appendix G, with references from [5-10]. Note that many SOTA performances are based on different classifiers, so these results should only serve as reference points and cannot be directly comparable with the performance of ClinGen and other baselines.
>
>
> > [6] Peng et al. "Transfer Learning in Biomedical Natural Language Processing: An Evaluation of BERT and ELMo on Ten Benchmarking Datasets." BioNLP 2019.
> >
> > [7] Tinn et al. "Fine-tuning large neural language models for biomedical natural language processing." Patterns 4.4 (2023).
> >
> > [8] Gu et al. "Domain-specific language model pretraining for biomedical natural language processing." Transactions on Computing for Healthcare 2021.
> >
> > [9] Yasunaga et al. "LinkBERT: Pretraining Language Models with Document Links." ACL 2022.
> >
> > [10] BLURB Leaderboard (https://microsoft.github.io/BLURB/leaderboard.html)
>
> ***
> > Q6: D_train in Equation 1 has only K (K<=5) samples per label?
>
> A: $D_{train}$ has only K samples per label, where we set K=5 in our experiments. We intentionally set K to a very small value to simulate a few-shot learning scenario. Recent work on language model fine-tuning suggests that it can help model to learn a good initialization before fine-tuning on synthetic samples [2].
>
> >[2] Meng et al. "Tuning language models as training data generators for augmentation-enhanced few-shot learning." ICML 2023.
>
> ***
> Thanks again for your feedback! Please let us know if you have any further questions.

---

> > ### Comment · Reviewer_CY6f · 2023-11-21
> >
> > Thanks for the clarification and responses! I have no further questions.

---

### Official Review · Reviewer_D87k · 2023-11-01

**Soundness:** 2 fair
**Presentation:** 3 good
**Contribution:** 2 fair
**Rating:** 3
**Confidence:** 4

**Summary:**

This paper proposes a new approach called CLINGEN for generating synthetic clinical text data using large language models (LLMs). The key idea is to leverage both external clinical knowledge graphs (KGs) and the knowledge encoded in LLMs to create informative and diverse prompts for guiding the LLM to generate high-quality and realistic synthetic data.

Specifically, CLINGEN extracts clinical topics from KGs and LLMs and writing style suggestions from LLMs. It composes this knowledge into prompts with a random topic and style to encourage diversity. The resulting synthetic data is used to train task-specific models.

The authors comprehensive benchmark experiments are conducted.

**Strengths:**

-The writing is clear and easy to undersand.
-Proposes a simple yet effective strategy to harness both structured KG knowledge and unstructured knowledge in LLMs to create informative prompts.
-Provides thorough empirical evaluation across diverse clinical tasks and datasets demonstrating consistent gains.

**Weaknesses:**

-The proposed prompting strategy is intuitive but lacking in novelty. Leveraging both KGs and LLMs is expected to help.
- To best of my knowledge, the generated data is still in-domain data, generating the style the same as the training data. So, it is not surprised  for me that using more data for training can improve model performance. It is more make sense to improve small model's over medical capacity , instead of overfit on a specific task with more training data.
-No rigorous comparison with other prompting optimization methods in the literature.

**Questions:**

- I am not sure why the two stages training is conducted. Was the results yo mix train your training data with  generated data together for training?

---

> ### Author Response · Authors · 2023-11-19
> **Initial Response to Reviewer D87k (Part 1)**
>
> We appreciate the time and detailed comments of the reviewer. We answer your questions as follows.
>
> ***
> > Q1: I am not sure why the two stages training is conducted. Was the results yo mix train your training data with generated data together for training?
>
> A: Thank you for the question. In Stage I, we perform fine-tuning of the classifier using few-shot examples in the original training set, specifically K samples per label (in our experiments, K=5). Subsequently, in Stage II, we conduct further fine-tuning of the classifier using the synthetic training data that we generate. We employ this two-stage training because it has been empirically demonstrated to be stable and is commonly employed in various works, including synthetic data generation tasks [1] and other semi-supervised learning methods [2,3]. Note that we mix the few-shot training data with the generated synthetic data and apply this two-stage training **consistently** for all the baselines to ensure a fair comparison.
>
> >[1] Meng et al. "Tuning language models as training data generators for augmentation-enhanced few-shot learning." ICML 2023
> >
> >[2] Laine and Aila. "Temporal ensembling for semi-supervised learning." ICLR 2017
> >
> >[3] Chen et al. "Mixtext: Linguistically-informed interpolation of hidden space for semi-supervised text classification." ACL 2020

---

> ### Author Response · Authors · 2023-11-19
> **Initial Response to Reviewer D87k (Part 2)**
>
> > W1: The proposed prompting strategy is intuitive but lacking in novelty. Leveraging both KGs and LLMs is expected to help.
>
> A: Thanks for the insightful feedback. We have answered this question in the general response for all reviewers. Please refer to the **Lack of novelty** section for details.
>
> ***
> > W1 (continued):  To best of my knowledge, the generated data is still in-domain data, generating the style the same as the training data. So, it is not surprised for me that using more data for training can improve model performance.
>
> A: We would like to clarify that we *do not* use “more data” compared to the baselines. In our experiments, all the baseline models are evaluated based on their respective generated synthetic datasets, and we maintain the **same amount of synthetic training data** across all methods, as mentioned in Appendix C. This ensures a fair comparison among all the methods evaluated in our study. In the revised version, we explicitly mention this in **Section 5.1** to avoid confusion.
>
> Moreover, it's essential to highlight that merely increasing the volume of training data **does not guarantee** better results. As shown in Table 1, some standard data augmentation methods cannot even outperform the supervised setting with few-shot examples only. Besides, for LLM-based data generation approaches, Figure 11(a) shows that the best baseline (DemoGen) using 100% of the training data actually shows lower performance compared to using only 50% of the training data. These observations illustrate the importance of the quality of training data over its quantity. Our focus, therefore, has been on developing a methodology to create high-quality synthetic training data. The results from our experiments in Table 1, Figure 5 and Figure 11 confirm that we consistently outperform baselines with the same amount of generated data under different budgets.
>
> ***
> > W1 (continued): It is more make sense to improve small model's over medical capacity, instead of overfit on a specific task with more training data.
>
> A: Thanks for the suggestion. We would like to emphasize that our primary goal is indeed to improve a model’s medical capacity during fine-tuning. Our approach focuses on enriching the training dataset with diverse medical knowledge, which can empower models of varying sizes with improved medical capacity. The generated synthetic data can also be easily and directly applied to improve various existing models without additional design.
>
> Besides, we believe that our approach (e.g., improving the data quality for fine-tuning task-specific models) is **orthogonal** to those efforts that improve the small models’ medical capacity in the pretraining stage. As shown in Table 1, the average performance of ClinGen with *PubMedBERT-base* (110M Parameters) as the backbone is *better than* the strongest baseline using *PubMedBERT-large* (340M Parameters) on the majority of tasks (e.g. Text Class, Relation Extraction, NLI, Fact Verification, and Medication Attribute Extraction), indicating that better data quality for fine-tuning is also crucial for adapting small model on specific tasks.
>
> Moreover, it is important to note that our strategy of incorporating more diverse training data alleviates rather than exacerbates the overfitting issue. By incorporating diverse training data enriched with medical knowledge, we enhance the ability of models to generalize across various clinical scenarios.
>
> ***
> > W2: No rigorous comparison with other prompting optimization methods in the literature.
>
> A: Thanks for the constructive feedback. We would like to highlight that the majority of existing research on synthetic data generation relies on manually crafted, straightforward prompts without dedicated designs [4,5,6]. That being said, we have tried our best to identify two closely related prompt optimization techniques that are potentially applicable to our case. Please see the **Comparison with different prompt designs** section of the general response to all reviewers for the experimental results and further discussions.
>
>
> > [4] Ye et al. "Zerogen: Efficient zero-shot learning via dataset generation." EMNLP 2022.
> >
> > [5] Meng et al. "Tuning language models as training data generators for augmentation-enhanced few-shot learning." ICML 2023.
> >
> > [6] Ye et al. "ProGen: Progressive zero-shot dataset generation via in-context feedback." EMNLP 2022.
>
> ***
> Thanks again for your constructive suggestions! We hope our responses address your concerns.

---

### Official Review · Reviewer_jB98 · 2023-11-03

**Soundness:** 3 good
**Presentation:** 3 good
**Contribution:** 2 fair
**Rating:** 6
**Confidence:** 4

**Summary:**

The paper aims to do synthetic clinical text generation using LLMs for clinical NLP tasks using the outlined approach -

(1) Extract clinical knowledge in the form of sample related entities or relations from external KG (nodes and edges from KB) or query relevant knowledge from LLM (prompt LLM to list 100 entities of a certain entity type).
(2) Include task names and  writing style into the prompt (e.g “medical literature” or “patient-doctor dialogues” ).
(3) Generate data using few-shot learning
(3) Use a pretrained model to fine tune on synthetic data.

The paper does a very comprehensive evaluation of  synthetic clinical data generation across 7 clinical NLP tasks and 16 datasets. Benchmarked with ZeroGen and DemoGen. Metrics included - evaluation task metrics, Entity Coverage. Entity Frequency, Central Moment Discrepancy (CMD), t-SNE  embeddings comparing synthetic and ground truth data. Includes a detailed ablation and parameter study.

**Strengths:**

The paper does a good job doing comprehensive evaluation and ablation study. The paper is well written.

**Weaknesses:**

To better assess the importance of the work, it would be beneficial to clarify the questions listed below.

**Questions:**

(1) The reviewer likes that the paper did a direct comparison with the prompts from GPT-3.5. However, I'm interested to know if the current medical LLMs are proficient in the tasks described. Should they already excel in these areas, it might somewhat diminish the necessity for the proposed approach.
(2) What are the limitations of your approach - what tasks is it limited to? How will it perform on more complex tasks such as descriptive QA (since most of the evaluation is on classification tasks)? Will it work on medical records?
(3) Some of the tasks still have some issues in regards to entity coverage and it looks like generally the performance is low on these tasks. Are these common entities that are missed and not identified by LLM or KB ? How will this looks like if a medical LLM was used for generation ?

---

> ### Author Response · Authors · 2023-11-19
> **Initial Response to Reviewer jB98 (Part 1)**
>
> We thank the reviewer for the time, valuable feedback, and suggestions for improvements. For your several questions, we clarify as follows:
>
> ***
> > Q1: The reviewer likes that the paper did a direct comparison with the prompts from GPT-3.5. However, I'm interested to know if the current medical LLMs are proficient in the tasks described. Should they already excel in these areas, it might somewhat diminish the necessity for the proposed approach.
>
> A: Thank you for the suggestion. Per your request, we have included additional experimental results that compare the direct inference performance of two recent and popular medical Large Language Models (LLMs): **PMC-LLaMa** [1] and **MedAlpaca** [2]. The table below illustrates that these medical LLMs perform less effectively than both ClinGen and GPT-3.5. This result can likely be attributed to two reasons:
> - The smaller size of the medical LLMs (7B) compared to GPT-3.5, which results in limited language modeling and reasoning capabilities. While we are aware that larger medical LLMs like Med-PaLM exist, they are not publicly accessible and thus cannot be used in our experiments.
> - Furthermore, these medical LLMs are typically trained on general medical knowledge sources, such as Wikipedia and Stack Exchange, as indicated by https://github.com/kbressem/medAlpaca#data-overview. This general knowledge may not translate effectively to specific tasks like chemical-protein interaction prediction or health fact verification, leading to suboptimal performance in few-shot learning scenarios.
>
> This observation aligns with findings from other research [3], which suggest that for certain clinical Natural Language Processing tasks, the fine-tuning of more compact models may outperform the in-context learning capabilities of larger LLMs. The results and discussions have been updated to **Table 2** in **Section 5.3**.
>
> | | HOC | | GAD | | ChemProt | MEDIQA-RQE | PUBHEALTH | | | NCBI-Disease | | | CASI | |
> |:--:|:--:|:--:|:--:|:--:|:--:|:--:|:--:|:--:|:--:|:--:|:--:|:--:|:--:|:--:|
> | | F1 | P | R | F1 | F1 | ACC | ACC | F1 | P | R | F1 | P | R | F1 |
> | GPT-3.5 Inference | 68.76 | 84.21 | **97.46** | 90.35 | 49.42 | 74.31 | **69.50** | **52.47** | 46.62 | 52.31 | 49.30 | 48.82 | **74.75** | 59.07 |
> | PMC-LLaMa Inference | 32.94 | 90.14 | 90.59 | 90.37 | 13.35 | 52.17 | 14.53 | 2.94 | 61.87 | 37.81 | 46.79 | 59.89 | 37.94 | 45.45 |
> | MedAlpaca Inference | 36.44 | 69.95 | 70.29 | 70.12 | 26.29 | 57.67 | 56.51 | 35.71 | 44.69 | 31.16 | 27.85 | 52.51 | 49.16 | 51.64 |
> | CLinGen w/ KG | 77.71 | 94.30 | 89.09 | **91.62** | 60.12 | **79.92** | 50.20 | 41.26 | **62.46** | **64.08** | **63.26** | 70.96 | 69.66 | **70.30** |
> | CLinGen w/ LLM | **78.14** | **95.08** | 86.14 | 90.39 | **63.05** | 77.36 | 52.96 | 43.31 | 61.12 | 60.16 | 60.64 | **71.61** | 66.86 | 69.15 |
>
>
> >[1] Wu et al. "Pmc-llama: Further finetuning llama on medical papers." arXiv preprint arXiv:2304.14454 (2023).
> >
> >[2] Han et al. "MedAlpaca--An Open-Source Collection of Medical Conversational AI Models and Training Data." arXiv preprint arXiv:2304.08247 (2023).
> >
> >[3] Lehman et al. "Do We Still Need Clinical Language Models?." CHIL 2023.
>
> *****
> > Q2: What are the limitations of your approach - what tasks is it limited to? How will it perform on more complex tasks such as descriptive QA (since most of the evaluation is on classification tasks)? Will it work on medical records?
>
> A: Thanks for the thoughtful feedback. We have answered this question in the general response for all reviewers. Please refer to the **Application to QA and EHR tasks** section of the general response for details.

---

> ### Author Response · Authors · 2023-11-19
> **Initial Response to Reviewer jB98 (Part 2)**
>
> > Q3: Some of the tasks still have some issues in regards to entity coverage and it looks like generally the performance is low on these tasks. Are these common entities that are missed and not identified by LLM or KB? How will this looks like if a medical LLM was used for generation?
>
> A: We appreciate your observation. We acknowledge that there remains a disparity in entity coverage between our generated training data and the ground truth data specifically on the LitCovid and CHEMDNER datasets.  This disparity is primarily due to the extensive and diverse sources of entities present in these datasets. Specifically, the LitCovid corpus and the CHEMDNER corpus include 24,960 and 10,000 PubMed articles, respectively, while the BC5CDR dataset only covers 500 PubMed articles. As a result, we observe that there exist more diverse entities for the original LitCovid and CHEMDNER training data. The wide range of entities present in LitCovid and CHEMDNER can be challenging to be fully covered using KGs or querying LLMs.
>
> Nevertheless, it's worth noting that despite this disparity with the ground truth, our approach **consistently outperforms or matches** baselines in both entity coverage and downstream model performance. We also attempted to utilize a medical LLM (MedAlpaca) for topic generation, but the results were not promising. We provide its entity coverage and downstream model performance results below:
>
> | Source of Topics| LitCovid | CHEMDNER |
> |:--:|:--:|:--:|
> | # unique entities per sample |||
> | KG | **0.307** | **0.402** |
> | LLM (ChatGPT) | 0.253 | 0.267 |
> | LLM (MedAlpaca) | 0.244 | 0.186 |
>
> |Models| LitCovid | CHEMDNER |
> |:--:|:--:|:--:|
> | PubMedBERT-Base | F1 | F1 |
> | ClinGen w/ KG | 58.01 | **56.94** |
> | ClinGen w/ LLM (ChatGPT) | **59.22** | 54.84 |
> | ClinGen w/ LLM (MedAlpaca) | 55.45 | 52.15 |
> | PubMedBERT-Large |||
> | ClinGen w/ KG | 55.81 | **55.56** |
> | ClinGen w/ LLM (ChatGPT) | **57.07** | 55.37 |
> | ClinGen w/ LLM (MedAlpaca) | 53.90 | 52.67 |
>
> The experimental results show that the topics generated by a medical LLM do not cover a greater number of clinical entities compared to our approach, and they also exhibit lower downstream performance. This could be attributed to the medical LLMs having fewer parameters than ChatGPT, which results in limited instruction-following capabilities. Upon inspecting the topics generated by MedAlpaca, we observe a **lack of diversity** among the entities, which may also contribute to their inferior performance. We provide 20 examples of these generated entities below:
>
> ```
> 1. Acebutolol 2. Acetaminophen 3. Acetylcysteine 4. Acetylsalicylic acid 5. Aciclovir 6. Acid 7. Acid chloride 8. Acid anhydrous 9. Acid citrate 10. Acid nitrate 11. Acid phosphate 12. Acid sulfate 13. Acidic 14. Acidity 15. Acidity regulator 16. Acidity regulator citric acid 17. Acidity regulator phosphoric acid 18. Acidity regulator sulfuric acid 19. Acidity regulator trichloroacetic acid 20. Acidity regulator trifluoroacetic acid
> ```
>
> One potential strategy to address this issue could involve collecting more task-specific labeled/unlabeled data so that we have more information about target distribution. However, this approach could deviate from the true few-shot scenario that we are investigating in this study [4]. For this reason, we have not included this method in the current explorations. Investigating effective ways to collect and utilize such data represents a valuable direction for future research.
>
>
> >[4] Perez et al. "True few-shot learning with language models." NeurIPS 2021.
>
> ***
> Thank you again for your review! We hope our responses can address your concerns. Please let us know if you have any further questions.

---

### Author Response · Authors · 2023-11-19
**General Response to All Reviewers (Part 1)**

We sincerely thank all reviewers for the insightful feedback. Your expert knowledge has helped us strengthen the manuscript.  Here, we provide a response to some general questions:

***
**Application to QA and EHR tasks (Reviewer jB98, CY6f):**

The primary goal of our research is to investigate the capabilities of LLMs in generating synthetic clinical text data for downstream classification tasks. We believe that this approach has broad applications where LLMs can offer significant advantages. To demonstrate this, we conduct comprehensive experiments specifically focused on a broad range of clinical NLP tasks, which cover 7 tasks across 16 datasets.

While we fully acknowledge the significance of other types of tasks and data, such as Question Answering (QA) tasks and structured Electronic Health Record (EHR) data, it's important to note that they present unique challenges. Therefore, these directions go beyond the scope of our current research, and we view them as potential areas for future exploration.

That being said, we conducted additional experiments on two multi-choice QA datasets, namely BioASQ [1] and PubMedQA [2] on ClinGen and the most relevant baselines. As presented in the table below, ClinGen consistently outperforms the baselines on these two datasets. However, when it comes to more intricate QA tasks designed to generate free-form answers, there exist inherent difficulties in evaluation, as it is challenging to establish quantitative metrics that reliably correlate with human accuracy judgments [3], even with the assistance of state-of-the-art LLMs [4]. Usually, human evaluators are required to assess the answer quality, which introduces subjectivity and scalability issues.

|| BioASQ | PubMedQA |
|:--:|:--:|:--:|
| PubMedBERT-Base | ACC | ACC |
| ZeroGen | 64.57 | 52.68 |
| ProGen | 65.71 | 54.83 |
| DemoGen | 62.71 | 54.65 |
| ClinGen w/ KG | 66.44 | 56.85 |
| ClinGen w/ LLM | **67.14** | **57.52** |
| PubMedBERT-Large |||
| ZeroGen | 68.66 | 55.05 |
| ProGen | 67.42 | 58.87 |
| DemoGen | 68.26 | 55.28 |
| ClinGen w/ KG | 69.25 | **61.79** |
| ClinGen w/ LLM | **70.71** | 61.12 |

On the other hand, EHR data falls within a distinct modality (i.e., tabular data) from textual data, which may require different methodologies and approaches [5]. Nonetheless, we are aware of the capabilities of LLMs in this context. Recent studies [6,7] have explored transforming tabular data into text to harness the power of LLMs, which yields promising results and shows the potential of LLMs for structured data generation. However, it's important to highlight that these approaches are fundamentally different from the methods we propose in this paper and are somehow beyond the scope of this paper.

We appreciate the reviewers's feedback and suggestions, and have revised the manuscript to add more discussion about this issue. Please refer to **Appendix A and K** for details.

>[1] Tsatsaronis et al. "An overview of the BIOASQ large-scale biomedical semantic indexing and question answering competition." BMC bioinformatics 2015
>
> [2] Jin et al. "PubMedQA: A Dataset for Biomedical Research Question Answering." EMNLP 2019
>
>[3] Chen et al. "Evaluating question answering evaluation." MRQA workshop 2019
>
>[4] Chen et al. "Exploring the use of large language models for reference-free text quality evaluation: A preliminary empirical study." arXiv 2023
>
>[5] Wornow et al. "The shaky foundations of large language models and foundation models for electronic health records." npj Digital Medicine 2023
>
>[6] Hegselmann et al. "Tabllm: Few-shot classification of tabular data with large language models." AISTATS 2023
>
>[7] Borisov et al. "Language models are realistic tabular data generators." ICLR 2023

***
**Patient privacy concerns (Reviewer CY6f, zJ5m):**

We thank the reviewers for raising the question regarding the patient privacy issue. We are well aware of this concern in clinical NLP. Specifically, we have carefully curated the five few-shot demonstrations to ensure they only contain conceptual information and are fully free from any Protected Health Information (PHI) related to patients.  With the five de-identified examples as the only data input for demonstrations, the synthetic training data we generate is highly *unlikely to include any private information*.

We also acknowledge the possibility of inadvertently introducing sensitive data through the GPT model itself. To address this, we have made a deliberate effort to avoid any instructions that can potentially extract sensitive patient information within the prompts. Instead, the prompts we used focus solely on obtaining conceptual information relevant to the target task. Lastly, we have conducted rigorous inspections of the generated synthetic data across all covered tasks to affirm that no such private information exists in the synthetic data generated by our method. We have included an additional paragraph in **Appendix A** to address this issue.

---

> ### Author Response · Authors · 2023-11-19
> **General Response to All Reviewers (Part 2)**
>
> **Comparison with different prompt designs (Reviewer D87k, CY6f):**
>
> We appreciate the helpful suggestions on adding additional prompt optimization comparison. For this purpose, we have carried out an additional analysis with two recent and representative prompt optimization techniques, namely **Reframe** [8] and **APE** [9]. In our setting, Reframe incorporates several principles (e.g., using low-level patterns, itemizing instructions, etc.) to produce high-quality prompts to enhance text generation, whereas APE leverages the LLM itself to automatically optimize the prompts based on the target task information. We demonstrate their performance on various clinical tasks in the following table.
>
> The results indicate that our proposed ClinGen consistently outperforms both baselines. This performance gain may be attributed to the fact that the prompts generated by Reframe and APE mainly focus on incorporating and decomposing task-specific information but do not *adequately address* the unique challenges for the clinical data generation task, i.e., distribution shift and lack of diversity.
>
> We have included the detailed experimental results and the designs of the prompts they proposed in **Appendix L** for further reference.
>
> || LitCovid | CDR | MEDIQA-RQE | MQP | CHEMDNER | BC5CDR-D | Average |
> |:--:|:--:|:--:|:--:|:--:|:--:|:--:|:--:|
> | PubMedBERT-Base | F1 | F1 | ACC | ACC | F1 | F1 ||
> | Reframe | 56.74 | 57.27 | 61.92 | 67.60 | 54.61 | 59.17 | 59.55 |
> | APE | 56.24 | 61.12 | 66.55 | 68.00 | 52.10 | 58.79 | 60.47 |
> | ClinGen w/ KG | 58.01 | 61.75 | **74.85** | 72.17 | **56.94** | 60.75 | **64.08** |
> | ClinGen w/ LLM | **59.22** | **63.34** | 72.40 | **73.31** | 54.84 | **61.03** | 64.02 |
> | PubMedBERT-Large ||||||||
> | Reframe | 54.06 | 58.78 | 66.57 | 71.30 | 55.05 | 60.41 | 61.03 |
> | APE | 53.54 | 61.65 | 69.20 | 71.00 | 53.03 | 59.87 | 61.38 |
> | ClinGen w/ KG | 55.81 | 62.66 | **79.92** | 75.82 | **55.56** | 61.21 | 65.16 |
> | ClinGen w/ LLM | **57.07** | **64.99** | 77.36 | **76.21** | 55.37 | **63.15** | **65.69** |
>
> >[8] Mishra, Swaroop, et al. "Reframing Instructional Prompts to GPTk's Language." ACL 2022
> >
> >[9] Zhou, Yongchao, et al. "Large language models are human-level prompt engineers." ICLR 2023.
>
> *****
> **Lack of novelty (Reviewer D87k, CY6f):**
>
> We thank the reviewers for the valuable feedback. ClinGen is rationally designed to address our identified issues of **limited domain knowledge coverage** and **distribution shifts** using KGs/LLMs, which have not been fully solved by existing works. We consider simplicity and effectiveness as one strength of our approach. Simplicity often leads to better generalizability, and our method has indeed led to consistent performance gain across 16 datasets covering 7 clinical NLP tasks. We also hope that our intuitive methodology will inspire further research into clinical text data generation using LLM-prompting techniques.
>
> While we have experimented with more complex prompt designs as detailed in papers [10,11], these designs do not lead to better performance when compared to ClinGen. Please see the **Comparison with different prompt designs** section of the general response for all reviewers for details.
>
> Furthermore, it is worth noting that we view our contribution as identifying the intrinsic challenges for LLM-based data generation on clinical NLP tasks (Section 3.2), and leveraging proper techniques to tackle them (Section 4). Although the basic idea of combining KGs and LLMs to benefit data generation is conceptually simple, the actual development of a concrete and effective approach to utilize these resources for performance enhancement is nontrivial.
> ClinGen is not a simple combination of KGs and LLMs, but an elaborate design to mitigate the identified drawbacks. As demonstrated in experiments, our design effectively overcomes these challenges (Section 6), resulting in notable performance gains (Table 1).
>
> >[10] Mishra, Swaroop, et al. "Reframing Instructional Prompts to GPTk's Language." ACL 2022
> >
> >[11] Zhou, Yongchao, et al. "Large language models are human-level prompt engineers." ICLR 2023.
>
> *****
> Thanks to all the reviewers again! We hope our responses have addressed your concerns. Please let us know if you have any further questions, and we are happy to discuss further.

---

### Author Response · Authors · 2023-11-19
**Summary of Revisions**

In response to valuable feedback from reviewers, we have diligently revised our manuscript, denoting these changes with $\textbf{\color{blue}blue}$ highlights. While answering detailed questions of each review, we also list a summary of the updates:
- We have added a comparison with direct inference of two medical LLMs in Table 2 in Section 5.3.
- We have incorporated the sota performance of “Supervised - Full” and the results of combining topics from KG and LLM in Table 6,7,8 in Appendix G for reference.
- We have added more discussion of limitations and future works in Appendix A.
- We have added experimental results of prompt optimization baselines and additional tasks (QA) in Appendix L and K, respectively.
- We clarify the scope of clinical knowledge, details of baselines, missing references, and experiment setups in Section 1, Appendix E, Appendix A, and Section 5.1, respectively.

We hope that the revised manuscript will adequately address the concerns raised by the reviewers.

---

### Author Response · Authors · 2023-11-21
**A Gentle Reminder**

Dear Reviewers,


Thank you again for your valuable feedback. We would like to kindly remind you that the author/reviewer discussion phase ends in 1 day, on November 22nd. We sincerely hope that our responses and the supplementary experiments have enhanced the paper's quality and addressed your concerns. If there are any additional suggestions or comments you would like to provide, please don't hesitate to share them. We look forward to engaging in a constructive discussion during the rebuttal phase.


Best Regards,

Submission #8883 Authors

---

### Meta-Review · Area_Chair_JJ8i · 2023-12-04

**Metareview:**

This paper addresses the problem of data augmentation in clinical text tasks. The main idea is to integrate knowledge from Knowledge Graphs (KGs) and Large Language Models (LLMs) into the prompts that are then used to generate synthetic data. This integration aims to generate synthetic data that is not only diverse but also considers distribution shift. As multiple reviewers pointed out,  the prompting method, which is the contribution of the work,  in its current form, offers limited new insights to both research and development communities. To strengthen the paper's impact, it could be useful to compare to existing discrete and continuous prompt optimization methods.

**Justification For Why Not Higher Score:**

The ideas have merit, but the execution of the ideas has not yet led to a contribution that is at the level of a top-tier conference.

**Justification For Why Not Lower Score:**

N/A

---

### Decision · Program_Chairs · 2024-01-16

Reject